# Private Edge Density Estimation for Random Graphs: Optimal, Efficient and Robust

**Hongjie Chen**
ETH Zürich

**Jingqiu Ding**
ETH Zürich

**Yiding Hua**
ETH Zürich

**David Steurer**
ETH Zürich

## Abstract

We give the first polynomial-time, differentially node-private, and robust algorithm for estimating the edge density of Erdős-Rényi random graphs and their generalization, inhomogeneous random graphs. We further prove information-theoretical lower bounds, showing that the error rate of our algorithm is optimal up to logarithmic factors. Previous algorithms incur either exponential running time or suboptimal error rates.

Two key ingredients of our algorithm are (1) a new sum-of-squares algorithm for robust edge density estimation, and (2) the reduction from privacy to robustness based on sum-of-squares exponential mechanisms due to Hopkins et al. (STOC 2023).

## 1 Introduction

Privacy has nowadays become a major concern in large-scale data processing. Releasing seemingly harmless statistics of a dataset could unexpectedly leak sensitive information of individuals (see e.g. [NS09, DSSU17] for privacy attacks). Differential privacy (DP) [DMNS06] has emerged as a by-now standard technique for protecting the privacy of individuals with rigorous guarantees. An algorithm is said to be differentially private if the distribution of its output remains largely unchanged under the change of a single data point in the dataset.

For datasets represented by graphs (e.g. social networks), two notions of differential privacy have been investigated in the literature: edge differential privacy [NRS07, KRSY11], where each edge is regarded as a data point; and node differential privacy [BBDS13, KNRS13], where each node along with its incident edges is regarded as a data point. Node differential privacy is an arguably more desirable notion than edge differential privacy. On the other hand, node differential privacy is also in general more difficult to achieve without compromising on utility, as many graph statistics usually have high sensitivity in the worst case. It turns out that many graph statistics can have significantly smaller sensitivity on typical graphs under natural distributional assumptions. Several recent works could thus manage to achieve optimal or nearly-optimal utility guarantees in a number of random graph parameter estimation problems [BCS15, BCSZ18, SU19, CDd+24].

In this paper, we continue this line of work and study perhaps the most elementary statistical task in graph data analysis: Given an $n$-node Erdős-Rényi random graph of which each edge is present with probability $p^\circ$ independently, output an estimate $\hat{p}$ of the edge density parameter $p^\circ$, subject to node differential privacy. We consider the error metric $|\hat{p}/p^\circ - 1|$ which can reflect the fact that, the task is more difficult for smaller $p^\circ$.

38th Conference on Neural Information Processing Systems (NeurIPS 2024).

Without privacy requirement, the empirical edge density[1] $\hat{p}$ achieves the information theoretically optimal error rate $|\hat{p}/p^\circ - 1| \leqslant \tilde{O}(1/(n\sqrt{p^\circ}))$. The standard way to achieve $\varepsilon$-differential node privacy is to add Laplace noise with standard deviation $\Theta(1/(\varepsilon n))$ to the empirical edge density $\hat{p}$. This will incur an additional privacy cost of $\Theta(1/(\varepsilon n p^\circ))$ which dominates the non-private error $\tilde{O}(1/(n\sqrt{p^\circ}))$. Surprisingly, Borgs et al. [BCSZ18] gave an algorithm with privacy cost only $\tilde{O}(1/(\varepsilon n\sqrt{np^\circ}))$ which is negligible to the non-private error for any $\varepsilon \gg 1/\sqrt{n}$. However, their algorithm is based on a general Lipschitz extension technique that has exponential running time. Later, Sealfon and Ullman [SU19] provided a polynomial-time algorithm based on smooth sensitivity with privacy cost $\tilde{O}(1/(\varepsilon n\sqrt{np^\circ}) + 1/(\varepsilon^2 n^2 p^\circ))$, which is much greater than that of [BCSZ18] for $\varepsilon \ll 1/(\sqrt{np^\circ})$. Moreover, [SU19] gives evidence that their approach is inherently prohibited from achieving better privacy cost. On the other hand, known lower bounds in [BCSZ18, SU19] are not for Erdős-Rényi random graphs. This leads us to the following question:

*Is there a polynomial-time, differentially node-private, and rate-optimal edge density estimation algorithm for Erdős-Rényi random graphs?*

We essentially settled this question in this paper. Specifically, we give a polynomial-time and differentially node-private algorithm with privacy cost $\tilde{O}(1/(\varepsilon n\sqrt{np^\circ}))$. Moreover, we show this error rate is optimal up to a logarithmic factor by proving an information-theoretical lower bound of $\Omega(1/(\varepsilon n\sqrt{np^\circ}))$. Our algorithm actually works for the more general inhomogeneous random graphs [BJR07]. The inhomogeneous random graph model encompasses any random graph model where edges appear independently (after conditioning on node labels). Notable examples include the stochastic block model [HLL83], the latent space model [HRH02], and graphon [BC17].

Our algorithm largely exploits the close connection between differential privacy and adversarial robustness in statistics. This connection dates back to [DL09] and has witnessed significant progress in the past few years [LKKO21, LKO22, KMV22, GH22, AUZ23, HKM22, HKMN23, AKT+23, CCAd+23, CDd+24]. In particular, a very recent line of works [HKM22, HKMN23, CDd+24] could efficiently achieve optimal or nearly-optimal accuracy guarantees in a number of high-dimensional statistical tasks, by integrating two powerful tools — sum-of-squares method [RSS18] and exponential mechanisms [MT07]— in robustness and privacy respectively. Our algorithm extends this line of work. The key technical ingredients of our algorithm are (1) a new sum-of-squares algorithm for robust edge density estimation and (2) an exponential mechanism whose score function is based on the sum-of-squares program. As a consequence, our private algorithm is also robust to adversarial corruptions.

## 1.1 Results

To state our results formally, we need the following definitions.

**Definition 1.1** (Node distance, neighboring graphs). Let $n \in \mathbb{N}$. The node distance between two $n$-node graphs $G$ and $G'$, denoted by $\text{dist}(G, G')$, is the minimum number of nodes in $G$ that need to be rewired to obtain $G'$. Moreover, we say $G$ and $G'$ are neighboring graphs if $\text{dist}(G, G') \leqslant 1$.

**Definition 1.2** (Node differential privacy). Let $\mathcal{G}$ be the set of graphs. A randomized algorithm $\mathcal{A} : \mathcal{G} \to \mathbb{R}$ is $\varepsilon$-differentially (node-)private if for every neighboring graphs $G, G'$ and every $S \subseteq \mathbb{R}$, we have

$$\mathbb{P}[\mathcal{A}(G) \in S] \leqslant e^\varepsilon \cdot \mathbb{P}[\mathcal{A}(G') \in S].$$

**Definition 1.3** (Node corruption model). Let $n \in \mathbb{N}$ and $\eta \in [0, 1]$. For an $n$-node graph $G$, we say an $n$-node graph $G'$ is an $\eta$-corrupted version of $G$ if $\text{dist}(G, G') \leqslant \eta n$.

**Erdős-Rényi random graphs.** We provide a polynomial-time, differentially node-private and robust edge density estimation algorithm for Erdős-Rényi random graphs.

---

[1]The (empirical) edge density of an $n$-node graph equals the number of edges divided by $n(n-1)/2$.

**Theorem 1.4** (Erdős-Rényi random graphs, combination of Theorem D.1 and Theorem F.1).
*There are constants $C_1, C_2, C_3$ such that the following holds. For any $\eta \leq C_1$, $\varepsilon \geq C_2 \log(n)/n$, and $p^\circ \geq C_3/n$, there exists a polynomial-time $\varepsilon$-differentially node-private algorithm which, given an $\eta$-corrupted Erdős-Rényi random graph $\mathbb{G}(n, p^\circ)$, outputs an estimate $\tilde{p}$ satisfying*

$$\left| \frac{\tilde{p}}{p^\circ} - 1 \right| \leq O\left( \frac{\sqrt{\log n}}{n\sqrt{p^\circ}} + \frac{\log^2 n}{\varepsilon n \sqrt{np^\circ}} + \frac{\eta \log n}{\sqrt{np^\circ}} \right),$$

*with probability $1 - n^{-\Omega(1)}$.*

The first term $O(\sqrt{\log n}/(n\sqrt{p^\circ}))$ is the sampling error that is necessary even without privacy or robustness. The second term $O(\log^2(n)/(\varepsilon n \sqrt{np^\circ}))$ is the privacy cost of our algorithm, which matches the exponential-time algorithm in [BCSZ18]. The third term $O(\eta \log n/\sqrt{np^\circ})$ is the robustness cost of our algorithm, which matches the information-theoretical lower bound $\Omega(\eta/\sqrt{np^\circ})$ in [AJK$^+$22, Theorem 1.5] up to a $\log n$ factor.

Moreover, we provide the following lower bound which shows that the privacy cost of our algorithm is optimal up to a $\log n$ factor.[2]

**Theorem 1.5** (Privacy lower bound for Erdős-Rényi random graphs). *Suppose there is an $\varepsilon$-differentially node-private algorithm that, given an Erdős-Rényi random graph $\mathbb{G}(n, p^\circ)$, outputs an estimate $\tilde{p}$ satisfying $|\tilde{p}/p^\circ - 1| \leq \alpha$ with probability $1 - \beta$. Then we must have*

$$\alpha \geq \Omega\left( \frac{\log(1/\beta)}{\varepsilon n \sqrt{np^\circ}} \right).$$

**Inhomogeneous random graphs.** Given an $n$-by-$n$ edge connection probability matrix $Q^\circ$, the inhomogeneous random graph model $\mathbb{G}(n, Q^\circ)$ defines a distribution over $n$-node graphs where each edge $\{i, j\}$ is present with probability $(Q^\circ)_{ij}$ independently.

We provide a polynomial-time, differentially node-private and robust edge density estimation algorithm for inhomogeneous random graphs.

**Theorem 1.6** (Inhomogeneous random graphs, combination of Theorem D.1 and Theorem E.1). *Let $Q^\circ$ be an $n$-by-$n$ edge connection probability matrix and let $p^\circ := \sum_{i,j} Q^\circ_{ij}/(n^2 - n)$. Suppose $\|Q^\circ\|_\infty \leq Rp^\circ$ for some $R$. There is a sufficiently small constant $c$ such that the following holds. For any $\eta$ such that $\eta \log(1/\eta)R \leq c$, there exists a polynomial-time $\varepsilon$-differentially node-private algorithm which, given an $\eta$-corrupted inhomogeneous random graph $\mathbb{G}(n, Q^\circ)$, outputs an estimate $\tilde{p}$ satisfying*

$$\left| \frac{\tilde{p}}{p^\circ} - 1 \right| \leq O\left( \frac{\sqrt{\log n}}{n\sqrt{p^\circ}} + \frac{R \log^2 n}{\varepsilon n} + R\eta \log(1/\eta) \right),$$

*with probability $1 - n^{-\Omega(1)}$.*

We improve on the previous private edge density estimation algorithm for inhomogeneous random graphs by Chen et al. [CDd$^+$24, Lemma 4.10]. Their algorithm is based on [SU19] and has privacy cost $\tilde{O}(R/(\varepsilon n) + 1/(\varepsilon^2 nd^\circ))$, while our algorithm only has privacy cost $\tilde{O}(R/(\varepsilon n))$. To the best of our knowledge, even without privacy requirement and in the special case of Erdős-Rényi random graphs, no previous algorithm can match our guarantees in the sparse regime. Specifically, when $d^\circ \ll \log n$ and $\eta \geq \Omega(1)$, our algorithm can provide a constant-factor approximation of $d^\circ$, while the best previous robust algorithm [AJK$^+$22] can not.

We also provide matching lower bounds, showing that the guarantee of our algorithm in Theorem 1.6 is optimal up to logarithmic factors.

---

[2]Borgs et al. [BCSZ18] proved a lower bound for a variant of Erdős-Rényi random graphs. However, it is not clear whether their proof technique can be easily extended to Erdős-Rényi random graphs.

**Theorem 1.7** (Robustness lower bound for inhomogeneous random graphs). *Suppose there is an algorithm satisfies the following guarantee for any symmetric matrix $Q^\circ \in [0,1]^{n \times n}$. Given an $\eta$-corrupted inhomogeneous random graph $\mathbb{G}(n, Q^\circ)$, the algorithm outputs an estimate $\hat{p}$ satisfying $|\hat{p}/p^\circ - 1| \leqslant \alpha$ with probability at least $0.99$, where $p^\circ = \sum_{i,j} Q^\circ_{ij}/(n^2 - n)$. Then we must have $\alpha \geqslant \Omega(R\eta)$, where $R = \max_{i,j} Q^\circ_{ij}/p^\circ$.*

**Theorem 1.8** (Privacy lower bound for inhomogeneous random graphs). *Suppose there is an $\varepsilon$-differentially node-private algorithm satisfies the following guarantee for any symmetric matrix $Q^\circ \in [0,1]^{n \times n}$. Given an inhomogeneous random graph $\mathbb{G}(n, Q^\circ)$, the algorithm outputs an estimate $\hat{p}$ satisfying $|\hat{p}/p^\circ - 1| \leqslant \alpha$ with probability $1 - \beta$, where $p^\circ = \sum_{i,j} Q^\circ_{ij}/(n^2 - n)$. Then we must have*

$$\alpha \geqslant \Omega\left(\frac{R \log(1/\beta)}{n\varepsilon}\right),$$

*where $R = \max_{i,j} Q^\circ_{ij}/p^\circ$.*

## 1.2 Techniques

We give an overview of the key techniques used to obtain our algorithm. As our techniques for Erdős-Rényi random graphs can be easily extended to the more general inhomogeneous random graph model, we will focus on Erdős-Rényi random graphs to avoid a proliferation of notation. Specifically, given an $\eta$-corrupted Erdős-Rényi random graph $\mathbb{G}(n, d^\circ/n)$, our goal is to output a private estimate of $d^\circ$.

**Reduction from privacy to robustness.** Hopkins et al. [HKMN23] and Asi et al. [AUZ23] independently discovered the following black-box reduction from privacy to robustness. Given a robust algorithm $\mathcal{A}_{\text{robust}}$, one can directly obtain a private algorithm via applying the exponential mechanism [MT07] with the following score function,

$$\text{score}(d; A) := \min_{A'}\left\{\text{dist}(A', A) \ : \ |\mathcal{A}_{\text{robust}}(A') - d| \leqslant 1/\text{poly}(n)\right\}, \tag{1.1}$$

where $A$ is the adjacency matrix of input graph and $d$ is a candidate estimate. For privacy analysis, note that the sensitivity of the above score function is bounded by 1, as the node distance between neighboring graphs is at most 1. For utility analysis, when the input graph is a typical Erdős-Rényi random graph, the exponential mechanism will with high probability output a $\hat{d}$ of score $O(\log(n)/\varepsilon)$. Then we can argue that such a $\hat{d}$ is close to $d^\circ$ using the robustness of $\mathcal{A}_{\text{robust}}$. For example, if we plug in the robust algorithm in [AJK+22, Theorem 1.3], then the corresponding exponential mechanism will only incur a privacy cost of $\tilde{O}(1/(\varepsilon n\sqrt{d^\circ}))$.

However, directly plugging in the robust algorithm in [AJK+22] will lead to an exponential-time algorithm, as a single evaluation of the score function requires enumerating all $n$-node graphs. To obtain a polynomial-time algorithm, we develop a new robust algorithm via the *sum-of-squares* method.[3]

**Robust algorithm via sum-of-squares.** The sum-of-squares method uses convex programming (in particular, semidefinite programming) to solve polynomial programming. It is a very powerful tool for designing polynomial-time robust estimators (see [RSS18]). To obtain a robust algorithm via sum-of-squares, we first identify a set of polynomial constraints that a typical (uncorrupted) Erdős-Rényi random graph would satisfy. Specifically, these polynomial constraints encode the following regularity conditions: (1) the degrees of the nodes are highly concentrated, and (2) the centered adjacency matrix is spectrally bounded.

---

[3]In general, the black-box reduction by [HKMN23, AUZ23] does not provide guarantees in terms of computational complexity. For the problem of robust edge density estimation under node corruption, there is no known sum-of-squares algorithm before our work, and we are only aware of the iterative algorithm [AJK+22]. For such algorithms not based on convex relaxation, it is completely unclear how to use the aforementioned connection between private and robust estimation towards an efficient private algorithm.

We also include the constraint that at most $\eta$ fraction of the nodes in the graph are corrupted. Then we give a proof that if a graph satisfies the above constraints, then its average degree will be close to $d^\circ$, even when $\eta$ fraction of nodes in the input graph are arbitrarily corrupted. Importantly, the proof is simple enough that it is captured by the sum-of-squares proof system (see [FKP+19]). This allows us to extend the utility guarantee of the polynomial program to its semidefinite programming relaxation, which results in a polynomial-time robust algorithm.

**Sum-of-squares exponential mechanism.** Given the above robust algorithm, we then use the sum-of-squares exponential mechanism developed in [HKM22, HKMN23] to obtain a private algorithm. More specifically, we apply the exponential mechanism with the sum-of-squares relaxation of the score function in Eq. (1.1). In this way, we obtain a private algorithm that is also robust to adversarial corruptions.

## 1.3 Notation

We introduce some notation used throughout this paper. We write $f \lesssim g$ to denote the inequality $f \leqslant C \cdot g$ for some absolute constant $C > 0$. We write $O(f)$ and $\Omega(f)$ to denote quantities $f_-$ and $f_+$ satisfying $f_- \lesssim f$ and $f \lesssim f_+$ respectively. We use boldface to denote random variables, e.g., $\boldsymbol{X}, \boldsymbol{Y}, \boldsymbol{Z}$. For a matrix $M$, we use $\|M\|_{\mathrm{op}}$ for the spectral norm of $M$. Let $\mathbb{1}$ and $\mathbb{0}$ denote the all-one and all-zero vector respectively, of which the size will be clear from the context. We use a graph $G$ and its adjacency matrix $A = A(G)$ interchangeably when there is no ambiguity. For an $n$-by-$n$ matrix $M$, we use $d(M)$ to denote its average row/column sum, i.e., $d(M) = \sum_{i,j} M_{ij}/n$. For any matrices (or vectors) $M, N$ of the same shape, we use $M \odot N$ to denote the element-wise product (aka Hadamard product) of $M$ and $N$.

## 1.4 Organization

The rest of the paper is organized as follows. In Section 2, we give a proof overview of our results and defer full proofs to the appendices. The appendices are organized as follows. We provide some sum-of-squares background in Appendix A and some concentration inequalities for random graphs in Appendix B. In Appendix C, we present a general sum-of-squares exponential mechanism that all of our private algorithms in this paper are based on. In Appendix D, we present our coarse estimation algorithm and give a full proof of its guarantees (Theorem D.1). In Appendix E, we present our fine estimation algorithm for inhomogeneous random graphs and give a full proof of its guarantees (Theorem E.1). In Appendix F, we present our fine estimation algorithm for Erdős-Rényi random graphs and give a full proof of its guarantees (Theorem F.1). All lower bounds are proved in Appendix G.

## 2 Private and robust algorithm for Erdős-Rényi random graphs

In this section, we describe our private and robust algorithm for Erdős-Rényi random graphs. We also give an overview of the analysis of our algorithm and sketch the proof of our lower bounds.

Our overall algorithm consists of two stages. In the first stage, we compute a coarse estimate that approximates the edge density parameter within constant factors. In the second stage, we improve the accuracy of this coarse estimate to the optimum. Since our algorithm is private in both stages, it is also private overall by the composition theorem of differential privacy (see [DR14, Section 3.5]).

We remark that for the Erdős-Rényi random graph model $\mathbb{G}(n, p^\circ)$, estimating its edge density parameter $p^\circ$ is equivalent to estimating its expected average degree $d^\circ := np^\circ$.[4] For the convenience of notation, we set our goal as estimating the expected average degree $d^\circ$ throughout this section.

---

[4]Strictly speaking, the expected average degree of $\mathbb{G}(n, p^\circ)$ should be $(n-1)p^\circ$. Here we call $np^\circ$ the expected average degree just for notational convenience. In the end, $(n-1)p^\circ = (1 - 1/n) \cdot np^\circ$.

## 2.1 General algorithm framework

Given an $n$-by-$n$ symmetric matrix $A$ and a scalar $\gamma \in [0,1]$, let $\mathcal{T}(Y, z; A, \gamma)$ be a polynomial system with indeterminates $Y = (Y_{ij})_{i,j\in[n]}$ and $z = (z_i)_{i\in[n]}$ that encodes the node distance between $Y$ and $A$:

$$\mathcal{T}(Y, z; A, \gamma) := \begin{cases} z \odot z = z, \ \langle \mathbb{1}, z \rangle \geq (1-\gamma)n \\ 0 \leq Y \leq \mathbb{1}\mathbb{1}^\top, \ Y = Y^\top \\ Y \odot zz^\top = A \odot zz^\top \end{cases}. \tag{2.1}$$

Let $\mathcal{R}(Y)$ be A polynomial system that encodes regularity conditions of Erdős-Rényi random graphs. The key observation here is that, for any $Y \in \{0,1\}^{n\times n}$ and $z \in \{0,1\}^n$ that satisfy constraints in $\mathcal{T}(Y, z; A, \gamma) \cup \mathcal{R}(Y)$, $Y$ is a graph that behaves like Erdős-Rényi random graphs (in the sense of the regularity conditions) and is within node distance $\gamma n$ to $A$ where they agree on $\{i \in [n] : z_i = 1\}$.

The key ingredient of our result is that, given proper regularity conditions $\mathcal{R}(Y)$, we can give degree-8 sum-of-squares proofs: for any $Y$ that satisfies constraints in $\mathcal{T}(Y, z; A, \gamma) \cup \mathcal{R}(Y)$, the average degree of $Y$ is close to the expected average degree $d^\circ$, even when the input graph $A$ is a $\gamma$-corrupted Erdős-Rényi random graph $\mathbb{G}(n, d^\circ/n)$. As a result of the sum-of-squares proofs-to-algorithms framework (see Theorem A.6), we can get an efficient and robust estimator $\tilde{\mathbb{E}}[d(Y)]$, where $\tilde{\mathbb{E}}$ is a pseudo-expectation obtained by solving level-8 sum-of-squares relaxation of $\mathcal{T}(Y, z; A, \gamma) \cup \mathcal{R}(Y)$.

Based on the above identifiability proof for robust estimation, we design a private and robust algorithm by applying the exponential mechanism[5] with the following score function:

$$\text{sos-score}(d; A) := \min_{0\leq\gamma\leq1} \gamma n \text{ s.t. } \exists \text{ level-8 pseudo-expectation } \tilde{\mathbb{E}} \text{ satisfying} \atop \mathcal{T}(Y, z; A, \gamma) \cup \mathcal{R}(Y) \cup \left\{|d(Y) - d| \leq 1/\text{poly}(n)\right\}. \tag{2.2}$$

Similar to Eq. (1.1), it is easy to observe this exponential mechanism is private.

**Lemma 2.1** (Privacy). *Consider the distribution $\mu_{A,\varepsilon}$ with support $[0, n]$ and density*

$$\mathrm{d}\mu_{A,\varepsilon}(d) \propto \exp(-\varepsilon \cdot \text{sos-score}(d; A)), \tag{2.3}$$

*where sos-score$(d; A)$ is defined in Eq. (2.2). A sample from $\mu_{A,\varepsilon}$ is $2\varepsilon$-differentially private.*

*Proof.* Since the node distance between neighboring graphs is at most 1, the sensitivity of the following score function is bounded by 1:

$$\text{score}(d; A) := \min_{0\leq\gamma\leq1} \gamma n \text{ s.t. } \mathcal{T}(Y, z; A, \gamma) \cup \mathcal{R}(Y) \cup \left\{|d(Y) - d| \leq 1/\text{poly}(n)\right\} \text{ is feasible.}$$

One can show that such sensitivity bound is inherited by its sum-of-squares relaxation sos-score as defined in Eq. (2.2). By a standard sensitivity-to-privacy argument (see e.g. [DR14, Theorem 3.10]), the exponential mechanism is $2\varepsilon$-differentially private. □

To analyze the utility of the private algorithm, we use the robustness of the score function. Assume the input graph is uncorrupted for simplicity. For a typical Erdős-Rényi random graph $A^\circ \sim \mathbb{G}(n, d^\circ/n)$, we have sos-score$(d^\circ, A^\circ) = 0$. By a standard volume argument (see e.g. [DR14, Theorem 3.11]), the exponential mechanism with high probability outputs a scalar $d$ satisfying sos-score$(d; A^\circ) \lesssim \log(n)/\varepsilon$. By the definition of our score function in Eq. (2.2), this implies that there exists a level-8 pseudo-distribution satisfying $\mathcal{T}(Y, z; A^\circ, \gamma) \cup \mathcal{R}(Y)$ with $\gamma \lesssim \log(n)/(\varepsilon n)$. The utility then follows from the above identifiability proof for robust estimation.

---

[5]To efficiently implement this exponential mechanism, we note that the score function Eq. (2.2) can be evaluated in polynomial time by combining binary search and semidefinite programming. By discretizing $[0, n]$ with step size $1/\text{poly}(n)$, one can sample from the distribution Eq. (2.3) with a polynomial number of queries to the score function. For more detailed discussions, see Remark C.1 and Remark C.2.

## 2.2 Coarse estimation

In this part, we describe a private and robust algorithm that can estimate the expected average degree $d^\circ$ within a constant approximation ratio.

**Theorem 2.2** (Coarse estimation algorithm, informal restatement of Theorem D.1). *For $\eta$ smaller than some constant, there is a polynomial-time $\varepsilon$-differentially node-private algorithm which, given an $\eta$-corrupted Erdős-Rényi random graph $\mathbb{G}(n, d^\circ/n)$, outputs an estimate $\hat{d}$ such that $|\hat{d} - d^\circ| \leqslant 0.5d^\circ$.*

We give a proof sketch of Theorem 2.2 at the end of this subsection. The formal theorem and proofs are deferred to Appendix D.

**Identifiability proof for robust estimation.** We first give a polynomial system that can identify the expected average degree $d^\circ$ up to constant factors, even when $\eta$-fraction of nodes are corrupted. Consider the following regularity condition on degrees:

$$\mathcal{R}(Y) := \left\{ (Y\mathbb{1})_i \leqslant 2\log(1/\eta) \cdot d(Y), \quad \forall i \in [n] \right\}. \tag{2.4}$$

The following lemma shows that Erdős-Rényi random graphs satisfy $\mathcal{T}(Y, z; A, 2\eta) \cup \mathcal{R}(Y)$ with high probability.

**Lemma 2.3** (Feasibility). *Let $A^\circ \sim \mathbb{G}(n, d^\circ/n)$ and let $A$ be an $\eta$-corrupted version of $A^\circ$. With high probability, there exists a graph $Y$ that satisfies the constraints in $\mathcal{T}(Y, z; A, 2\eta) \cup \mathcal{R}(Y)$.*

*Proof sketch.* For $d^\circ \gg \log(n)$, the maximum degree of $A^\circ$ is of order $O(d^\circ)$. Therefore, the uncorrupted graph $A^\circ$ satisfies the constraints. For $d^\circ \ll \log n$, using concentration properties of random graphs, we can show that the number of high degree nodes is bounded by $\eta n$. A feasible graph can then be obtained from the uncorrupted graph $A^\circ$ by trimming these highest degree nodes. □

Next, we show that these polynomial constraints give an identifiability proof for the expected average degree $d^\circ$.

**Lemma 2.4** (Identifiability). *Let $A^\circ \sim \mathbb{G}(n, d^\circ/n)$ and let $A$ be an $\eta$-corrupted version of $A^\circ$. For $\eta$ smaller than some constant and $\gamma \leqslant O(\eta)$, with high probability there is a degree-8 sum-of-squares proof that, if $Y$ satisfies $\mathcal{T}(Y, z; A, \gamma) \cup \mathcal{R}(Y)$, then $|d(Y) - d^\circ| \leqslant 0.001d^\circ$.*

*Proof sketch.* We first assume that $d^\circ \gg \log(n)$, for which the proof is simpler. By the degree-bound constraint $\mathcal{R}(Y)$, we have $n|d(Y) - d(A^\circ)| \leqslant 2\log(1/\eta) \cdot (d(Y) + d^\circ) \cdot \mathrm{dist}(Y, A^\circ)$. Using the constraints $Y \odot zz^\top = A \odot zz^\top$ and $\langle 1, z \rangle \geqslant (1 - \gamma)n$, we have $\mathrm{dist}(Y, A) \leqslant \gamma n$. Since $\mathrm{dist}(A, A^\circ) \leqslant \eta n$, by triangle inequality, we have $\mathrm{dist}(Y, A^\circ) \leqslant (\gamma + \eta)n$. Therefore, we have $|d(Y) - d(A^\circ)| \leqslant 0.0001d^\circ$ when $\gamma, \eta$ are at most some small constants. Finally, by random graph concentration, we have $|d^\circ - d(A^\circ)| \leqslant o(d^\circ)$ with high probability. Therefore, we have $|d(Y) - d(A^\circ)| \leqslant 0.001d^\circ$.

To deal with the sparse regime where $d^\circ \ll \log n$, we need to truncate the nodes of $A^\circ$ with degree $\Omega(\log(1/\eta)d^\circ)$. Our key observation is that, the average degree of the graph before and after truncation only differ by a constant factor. Therefore, we can still get $|d(Y) - d(A^\circ)| \leqslant 0.001d^\circ$.

Furthermore, it can be shown that this proof is a degree-8 sum-of-squares proof. □

**Robust algorithm via sum-of-squares.** Consider the algorithm that finds a level-8 pseudo-expectation satisfying $\mathcal{T}(Y, z; A, 2\eta) \cup \mathcal{R}(Y)$ —with $\mathcal{R}(Y)$ given in Eq. (2.4)— and outputs $\tilde{\mathbb{E}}[d(Y)]$. By Lemma 2.3, such a pseudo-expectation $\tilde{\mathbb{E}}$ exists with high probability. It follows from the sum-of-squares identifiability proof in Lemma 2.4 that $|\tilde{\mathbb{E}}[d(Y)] - d^\circ| \leqslant 0.001d^\circ$. Moreover, the algorithm can be implemented by semidefinite programming and run in polynomial time.

**Private and robust algorithm via sum-of-squares exponential mechanism.** We present our private and robust algorithm in Algorithm 2.5 and give a proof sketch of Theorem 2.2.

---

**Algorithm 2.5** (Private coarse estimation for Erdős-Rényi random graphs).
**Input:** $\eta$-corrupted Erdős-Rényi random graph $A$.

**Privacy parameter:** $\varepsilon$.

**Output:** A sample from the distribution $\mu_{A,\varepsilon}$ with support $[0, n]$ and density

$$\mathrm{d}\mu_{A,\varepsilon}(d) \propto \exp(-\varepsilon \cdot \text{sos-score}(d; A)), \tag{2.5}$$

where

$$\text{sos-score}(d; A) := \min_{0 \leqslant \gamma \leqslant 1} \gamma n \text{ s.t. } \exists \text{ level-8 pseudo-expectation } \tilde{\mathbb{E}} \text{ satisfying}$$
$$\mathcal{T}(Y, z; A, \gamma) \cup \mathcal{R}(Y) \cup \left\{|d(Y) - d| \leqslant 1/\text{poly}(n)\right\}, \tag{2.6}$$

with $\mathcal{R}(Y)$ given in Eq. (2.4).

---

*Proof sketch of Theorem 2.2. Privacy.* By Lemma 2.1, Algorithm 2.5 is $2\varepsilon$-differentially private.

*Utility.* For simplicity, we consider the case when there is no corruption (i.e. $\eta = 0$). The analysis for the case when $\eta > 0$ is similar. Let $A^\circ \sim \mathbb{G}(n, d^\circ/n)$. Then with high probability sos-score$(d^\circ; A^\circ) = 0$. By a standard volume argument, Algorithm 2.5 outputs a scalar $d$ that satisfies sos-score$(d; A^\circ) \lesssim \log(n)/\varepsilon$ with high probability. By the definition of sos-score in Eq. (2.6), this implies that there exists a level-8 pseudo-distribution satisfying $\mathcal{T}(Y, z; A, \gamma) \cup \mathcal{R}(Y) \cup \{|d(Y) - d| \leqslant 1/\text{poly}(n)\}$ with $\gamma \lesssim \log(n)/(\varepsilon n)$. When $\log(n)/(\varepsilon n)$ is at most a small constant, it follows from our sum-of-squares identifiability proof in Lemma 2.4 that, Algorithm 2.5 outputs a constant-factor approximation of $d^\circ$ with high probability. □

## 2.3 Fine estimation

From Section 2.2, we know how to obtain a constant-factor approximation of $d^\circ$ privately and robustly. In this section, we show how to improve the accuracy to the optimum.

**Theorem 2.6** (Fine estimation algorithm, informal restatement of Theorem F.1). *Let $0.5d^\circ \leqslant \hat{d} \leqslant 2d^\circ$. For $\eta$ smaller than some constant, there is a polynomial-time $\varepsilon$-differentially node-private algorithm which, given an $\eta$-corrupted Erdős-Rényi random graph $\mathbb{G}(n, d^\circ/n)$ and $\hat{d}$, outputs an estimate $\tilde{d}$ such that*

$$\left|\frac{\tilde{d}}{d^\circ} - 1\right| \leqslant \tilde{O}\left(\frac{1}{\sqrt{nd^\circ}} + \frac{1}{\varepsilon n \sqrt{d^\circ}} + \frac{\eta}{\sqrt{d^\circ}}\right).$$

We give a proof sketch of Theorem 2.6 at the end of this section. The formal theorem and proofs are deferred to Appendix F.

**Identifiability proof for robust estimation.** We first give a polynomial system which can identify the expected average degree $d^\circ$ with optimal error rate, when provided with a coarse estimate $\hat{d}$. Consider the following regularity conditions on degrees and eigenvalues:

$$\mathcal{R}(Y) := \left\{ \begin{array}{ll} |(Y\mathbb{1})_i - d(Y)| \leqslant \sqrt{\hat{d}} \log n, & \forall i \in [n] \\ \left\|Y - \frac{d(Y)}{n}\mathbb{1}\mathbb{1}^\top\right\|_{\text{op}} \leqslant \sqrt{\hat{d}} \log n & \end{array} \right\}. \tag{2.7}$$

**Lemma 2.7** (Feasibility). *Let $A^\circ \sim \mathbb{G}(n, d^\circ/n)$ and let $A$ be an $\eta$-corrupted version of $A^\circ$. Suppose $d^\circ/2 \leqslant \hat{d} \leqslant 2d^\circ$. Then with high probability, there exists a graph $Y$ that satisfies the constraints in $\mathcal{T}(Y, z; A, \eta) \cup \mathcal{R}(Y)$.*

*Proof.* By Chernoff bound, with high probability, the degree of each node in $A^\circ$ deviates from $d^\circ$ by at most $O(\sqrt{d^\circ} \log n)$. By the concentration of the spectral norm of random

matrices [BvH16], with high probability, we have $\|A^\circ - \frac{d(A^\circ)}{n}\mathbb{1}\mathbb{1}^\top\|_{op} \lesssim \sqrt{d^\circ \log n}$. Hence, $\mathcal{T}(Y, z; A, \eta) \cup \mathcal{R}(Y)$ is satisfied by $Y = A^\circ$ and $z = z^\circ$ where $z^\circ$ is the indicator vector for uncorrupted nodes. $\qquad\square$

Next we give a sum-of-squares identifiability proof for expected average degree estimation with optimal accuracy.

**Lemma 2.8** (Identifiability). *Let $A^\circ \sim \mathbb{G}(n, d^\circ/n)$ and let $A$ be an $\eta$-corrupted version of $A^\circ$. Suppose $d^\circ/2 \leqslant \hat{d} \leqslant 2d^\circ$. For $\eta$ smaller than some constant and $\gamma \leqslant O(\eta)$, with high probability there is a degree-8 sum-of-squares proof that, if $Y$ satisfies $\mathcal{T}(Y, z; A, \eta) \cup \mathcal{R}(Y)$, then*

$$\left| \frac{d(Y)}{d^\circ} - 1 \right| \leqslant \tilde{O}\left( \frac{1}{\sqrt{nd^\circ}} + \frac{\eta}{\sqrt{d^\circ}} \right).$$

*Proof sketch.* Let $Y_1, Y_2$ be two graphs satisfying the regularity condition $\mathcal{R}(Y_1)$ and $\mathcal{R}(Y_2)$ as described in Eq. (2.7), respectively. We give sum-of-squares proof that, if $\text{dist}(Y_1, Y_2) \leqslant \zeta n$ and $\zeta$ is at most some small constant, then $|d(Y_1) - d(Y_2)| \leqslant \zeta\sqrt{\hat{d}}\log n$ .

Let $w \in \{0, 1\}^n$ be the indicator vector for the shared induced subgraph between $Y_1$ and $Y_2$, i.e $Y_1 \odot ww^\top = Y_2 \odot ww^\top$. When $\text{dist}(Y_1, Y_2) \leqslant \zeta n$, we have $\langle w, \mathbb{1} \rangle \geqslant (1 - \zeta)n$. We have

$$
\begin{aligned}
n\Big(d(Y_1) - d(Y_2)\Big) &= \langle Y_1 - Y_2, \mathbb{1}\mathbb{1}^\top \rangle \\
&= \langle Y_1 - Y_2, \mathbb{1}\mathbb{1}^\top - ww^\top \rangle \\
&= \langle Y_1 - \frac{d(Y_1)}{n}\mathbb{1}\mathbb{1}^\top + \frac{d(Y_1)}{n}\mathbb{1}\mathbb{1}^\top - \frac{d(Y_2)}{n}\mathbb{1}\mathbb{1}^\top + \frac{d(Y_2)}{n}\mathbb{1}\mathbb{1}^\top - Y_2, \mathbb{1}\mathbb{1}^\top - ww^\top \rangle \\
&= \langle Y_1 - \frac{d(Y_1)}{n}\mathbb{1}\mathbb{1}^\top, \mathbb{1}\mathbb{1}^\top - ww^\top \rangle + \langle \frac{d(Y_2)}{n}\mathbb{1}\mathbb{1}^\top - Y_2, \mathbb{1}\mathbb{1}^\top - ww^\top \rangle \\
&\quad + \langle \frac{d(Y_1)}{n}\mathbb{1}\mathbb{1}^\top - \frac{d(Y_2)}{n}\mathbb{1}\mathbb{1}^\top, \mathbb{1}\mathbb{1}^\top - ww^\top \rangle .
\end{aligned}
$$

By rearranging terms, we can get

$$\frac{\langle \mathbb{1}, w \rangle^2}{n}\Big(d(Y_1) - d(Y_2)\Big) = \langle Y_1 - \frac{d(Y_1)}{n}\mathbb{1}\mathbb{1}^\top, \mathbb{1}\mathbb{1}^\top - ww^\top \rangle + \langle \frac{d(Y_2)}{n}\mathbb{1}\mathbb{1}^\top - Y_2, \mathbb{1}\mathbb{1}^\top - ww^\top \rangle .$$

For the first term $\langle Y_1 - \frac{d(Y_1)}{n}\mathbb{1}\mathbb{1}^\top, \mathbb{1}\mathbb{1}^\top - ww^\top \rangle$, we have

$$\langle Y_1 - \frac{d(Y_1)}{n}\mathbb{1}\mathbb{1}^\top, \mathbb{1}\mathbb{1}^\top - ww^\top \rangle = 2\langle Y_1 - \frac{d(Y_1)}{n}\mathbb{1}\mathbb{1}^\top, \mathbb{1}(\mathbb{1} - w)^\top \rangle + \langle \frac{d(Y_1)}{n}\mathbb{1}\mathbb{1}^\top - Y_1, (\mathbb{1} - w)(\mathbb{1} - w)^\top \rangle .$$

From constraints $|(Y_1\mathbb{1})_i - d(Y_1)| \leqslant \sqrt{\hat{d}}\log(n)$ for all $i \in [n]$, we have

$$\langle Y_1 - \frac{d(Y_1)}{n}\mathbb{1}\mathbb{1}^\top, \mathbb{1}(\mathbb{1} - w)^\top \rangle = \langle Y_1\mathbb{1} - d(Y_1)\mathbb{1}, \mathbb{1} - w \rangle \leqslant \zeta n \log(n)\sqrt{\hat{d}} .$$

From constraints $\left\| Y_1 - \frac{d(Y_1)}{n}\mathbb{1}\mathbb{1}^\top \right\|_{op} \leqslant \delta\sqrt{\hat{d}}$, we have

$$\langle \frac{d(Y_1)}{n}\mathbb{1}\mathbb{1}^\top - Y_1, (\mathbb{1} - w)(\mathbb{1} - w)^\top \rangle \leqslant \left\| Y_1 - \frac{d(Y_1)}{n}\mathbb{1}\mathbb{1}^\top \right\|_{op} \|\mathbb{1} - w\|_2^2 \leqslant \zeta n \sqrt{\hat{d}}\log(n) ,$$

The same bounds also apply for the second term $\langle Y_2 - \frac{d(Y_2)}{n}\mathbb{1}\mathbb{1}^\top, \mathbb{1}\mathbb{1}^\top - ww^\top \rangle$. Since $\langle \mathbb{1}, w \rangle \geqslant \Omega(n)$, it follows that $|d(Y_1) - d(Y_2)| \leqslant \tilde{O}(\zeta\sqrt{\hat{d}}) \leqslant \tilde{O}(\zeta\sqrt{d^\circ})$.

Since the original uncorrupted graph satisfies the regularity conditions, this gives the identifiability proof that $|d(Y) - d(A^\circ)| \leqslant \tilde{O}(\zeta\sqrt{d^\circ})$. By random graph concentration, with high probability, we have $|d^\circ - d(A^\circ)| \leqslant \tilde{O}(\sqrt{d^\circ/n})$. The claim thus follows. $\qquad\square$

**Robust algorithm via sum-of-squares.** Consider the algorithm that finds a level-8 pseudo-expectation satisfying $\mathcal{T}(Y, z; A, \eta) \cup \mathcal{R}(Y)$ —with $\mathcal{R}(Y)$ given in Eq. (2.7)— and outputs $\tilde{\mathbb{E}}[d(Y)]$. By Lemma 2.7, such a pseudo-expectation $\tilde{\mathbb{E}}$ exists with high probability. It follows from the sum-of-squares identifiability proof in Lemma 2.8 that $|\tilde{\mathbb{E}}[d(Y)]/d^\circ - 1| \leqslant \tilde{O}(1/\sqrt{nd^\circ} + \eta/\sqrt{d^\circ})$. Moreover, the algorithm can be implemented by semidefinite programming and run in polynomial time.

**Private and robust algorithm via sum-of-squares exponential mechanism.** We present our private and robust algorithm in Algorithm 2.9 and give a proof sketch of Theorem 2.6.

---

**Algorithm 2.9** (Private fine estimation for Erdős-Rényi random graphs)**.**

**Input:** $\eta$ corrupted random graph $A$, $\varepsilon$-differentially private coarse estimate $\hat{d}$.

**Privacy parameter:** $\varepsilon$.

**Output:** A sample from the distribution $\mu_{A,\varepsilon}$ with support $[0, n]$ and density

$$\mathrm{d}\mu_{A,\varepsilon}(d) \propto \exp(-\varepsilon \cdot \text{sos-score}(d; A)), \qquad (2.8)$$

where sos-score$(d; A)$ is defined as

$$\text{sos-score}(d; A) := \min_{0 \leqslant \gamma \leqslant 1} \gamma n \text{ s.t. } \exists \text{ level-8 pseudo-expectation } \tilde{\mathbb{E}} \text{ satisfying}$$
$$\mathcal{T}(Y, z; A, \gamma) \cup \mathcal{R}(Y) \cup \left\{ |d(Y) - d| \leqslant 1/\text{poly}(n) \right\}, \qquad (2.9)$$

with $\mathcal{R}(Y)$ given in Eq. (2.7).

---

*Proof sketch of Theorem 2.6. Privacy.* By Lemma 2.1, Algorithm 2.9 is $2\varepsilon$-differentially private. *Utility.* For simplicity, we consider the case when there is no corruption (i.e. $\eta = 0$). The analysis for the case when $\eta > 0$ is similar. Let $A^\circ \sim \mathbb{G}(n, d^\circ/n)$. Then with high probability sos-score$(d^\circ; A^\circ) = 0$. By a standard volume argument, Algorithm 2.9 outputs a scalar $d$ that satisfies sos-score$(d; A^\circ) \lesssim \log(n)/\varepsilon$ with high probability. By the definition of sos-score in Eq. (2.9), this implies that with high probability there exists a level-8 pseudo-distribution satisfying $\mathcal{T}(Y, z; A, \gamma) \cup \mathcal{R}(Y)$ with $\gamma \lesssim \log(n)/(\varepsilon n)$. Taking $\eta = \log(n)/(\varepsilon n)$ in Lemma 2.8, it follows that Algorithm 2.9 outputs an estimate $\tilde{d}$ such that $|\tilde{d}/d^\circ - 1| \leqslant \tilde{O}(1/\sqrt{nd^\circ} + 1/(\varepsilon n \sqrt{d^\circ}))$ with high probability. $\qquad\square$

## 2.4 Lower bound

We sketch the proof of Theorem 1.5. Let $\alpha \in [0, 1]$ and $d = (1 - \alpha)d^\circ$. We can construct a coupling $\omega$ between the distributions $\mathbb{G}(n, d/n)$ and $\mathbb{G}(n, d^\circ/n)$ with the following property. For $(G, G') \sim \omega$, we have dist$(G, G')$ bounded by $\tilde{O}(\alpha n \sqrt{d^\circ})$ with overwhelmingly high probability. By the definition of differential privacy, when $\varepsilon \alpha n \sqrt{d^\circ} \leqslant 1/\text{polylog}(n)$, the output of an $\varepsilon$-differentially private algorithm are indistinguishable under $\mathbb{G}(n, d/n)$ and $\mathbb{G}(n, d^\circ/n)$. Therefore, by setting $\alpha = \tilde{O}(1/\varepsilon n \sqrt{d^\circ})$, we conclude that no $\varepsilon$-differentially private algorithm can achieve error rate better than $\tilde{O}(1/\varepsilon n \sqrt{d^\circ})$. This provides a matching lower bound for our private edge density estimation algorithm.

## Acknowledgements

This project has received funding from the European Research Council (ERC) under the European Union's Horizon 2020 research and innovation programme (grant agreement No 815464). We thank the anonymous reviewers for constructive feedback.

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

# A  Sum-of-squares background

## A.1  Sum-of-squares hierarchy

In this paper, we use the sum-of-squares semidefinite programming hierarchy [BS14, BS16, RSS18] for both algorithm design and analysis. The sum-of-squares proof-to-algorithm framework has been proven useful in many optimal or state-of-the-art results in algorithmic statistics [HL18, KSS18, PS17, Hop20]. We provide here a brief introduction to pseudo-distributions, sum-of-squares proofs, and sum-of-squares algorithms.

**Pseudo-distribution.**   We can represent a finitely supported probability distribution over $\mathbb{R}^n$ by its probability mass function $\mu : \mathbb{R}^n \to \mathbb{R}$ such that $\mu \geq 0$ and $\sum_{x \in \text{supp}(\mu)} \mu(x) = 1$. We define pseudo-distributions as generalizations of such probability mass distributions by relaxing the constraint $\mu \geq 0$ to only require that $\mu$ passes certain low-degree non-negativity tests.

**Definition A.1** (Pseudo-distribution).  A *level-$\ell$ pseudo-distribution* $\mu$ over $\mathbb{R}^n$ is a finitely supported function $\mu : \mathbb{R}^n \to \mathbb{R}$ such that $\sum_{x \in \text{supp}(\mu)} \mu(x) = 1$ and $\sum_{x \in \text{supp}(\mu)} \mu(x) f(x)^2 \geq 0$ for every polynomial $f$ of degree at most $\ell/2$.

We can define the expectation of a pseudo-distribution in the same way as the expectation of a finitely supported probability distribution.

**Definition A.2** (Pseudo-expectation).  Given a pseudo-distribution $\mu$ over $\mathbb{R}^n$, we define the *pseudo-expectation* of a function $f : \mathbb{R}^n \to \mathbb{R}$ by

$$\tilde{\mathbb{E}}_{\mu} f := \sum_{x \in \text{supp}(\mu)} \mu(x) f(x) . \tag{A.1}$$

The following definition formalizes what it means for a pseudo-distribution to satisfy a system of polynomial constraints.

**Definition A.3** (Constrained pseudo-distributions).  Let $\mu : \mathbb{R}^n \to \mathbb{R}$ be a level-$\ell$ pseudo-distribution over $\mathbb{R}^n$. Let $\mathcal{A} = \{f_1 \geq 0, \dots, f_m \geq 0\}$ be a system of polynomial constraints. We say that $\mu$ *satisfies* $\mathcal{A}$ at level $r$, denoted by $\mu \mathrel{\vert\!\frac{}{r}} \mathcal{A}$, if for every multiset $S \subseteq [m]$ and every sum-of-squares polynomial $h$ such that $\deg(h) + \sum_{i \in S} \max\{\deg(f_i), r\} \leq \ell$,

$$\tilde{\mathbb{E}}_{\mu} h \cdot \prod_{i \in S} f_i \geq 0 . \tag{A.2}$$

We say $\mu$ satisfies $\mathcal{A}$ and write $\mu \models \mathcal{A}$ (without further specifying the degree) if $\mu \mathrel{\vert\!\frac{}{0}} \mathcal{A}$.

We remark that if $\mu$ is an actual finitely supported probability distribution, then we have $\mu \models \mathcal{A}$ if and only if $\mu$ is supported on solutions to $\mathcal{A}$.

**Sum-of-squares proof.**   We introduce sum-of-squares proofs as the dual objects of pseudo-distributions, which can be used to reason about properties of pseudo-distributions. We say a polynomial $p$ is a sum-of-squares polynomial if there exist polynomials $(q_i)$ such that $p = \sum_i q_i^2$.

**Definition A.4** (Sum-of-squares proof).  A *sum-of-squares proof* that a system of polynomial constraints $\mathcal{A} = \{f_1 \geq 0, \dots, f_m \geq 0\}$ implies $q \geq 0$ consists of sum-of-squares polynomials $(p_S)_{S \subseteq [m]}$ such that[6]

$$q = \sum_{\text{multiset } S \subseteq [m]} p_S \cdot \prod_{i \in S} f_i .$$

If such a proof exists, we say that $\mathcal{A}$ *(sos-)proves* $q \geq 0$ within degree $\ell$, denoted by $\mathcal{A} \mathrel{\vert\!\frac{}{\ell}} q \geq 0$. In order to clarify the variables quantified by the proof, we often write $\mathcal{A}(x) \mathrel{\vert\!\frac{x}{\ell}} q(x) \geq 0$.

---

[6]Here we follow the convention that $\prod_{i \in S} f_i = 1$ for $S = \emptyset$.

We say that the system $\mathcal{A}$ *sos-refuted* within degree $\ell$ if $\mathcal{A} \vdash_{\ell} -1 \geqslant 0$. Otherwise, we say that the system is *sos-consistent* up to degree $\ell$, which also means that there exists a level-$\ell$ pseudo-distribution satisfying the system.

The following lemma shows that sum-of-squares proofs allow us to deduce properties of pseudo-distributions that satisfy some constraints.

**Lemma A.5.** *Let $\mu$ be a pseudo-distribution, and let $\mathcal{A}, \mathcal{B}$ be systems of polynomial constraints. Suppose there exists a sum-of-squares proof $\mathcal{A} \vdash_{r'} \mathcal{B}$. If $\mu \models_r \mathcal{A}$, then $\mu \models_{r \cdot r' + r'} \mathcal{B}$.*

**Sum-of-squares algorithm.** Given a system of polynomial constraints, the *sum-of-squares algorithm* searches through the space of pseudo-distributions that satisfy this polynomial system by semidefinite programming.

Since semidefinite programing can only be solved approximately, we can only find pseudo-distributions that approximately satisfy a given polynomial system. We say that a level-$\ell$ pseudo-distribution *approximately satisfies* a polynomial system, if the inequalities in Eq. (A.2) are satisfied up to an additive error of $2^{-n^{\ell}} \cdot \|h\| \cdot \prod_{i \in S} \|f_i\|$, where $\|\cdot\|$ denotes the Euclidean norm[7] of the coefficients of a polynomial in the monomial basis.

**Theorem A.6** (Sum-of-squares algorithm). *There exists an $(n + m)^{O(\ell)}$-time algorithm that, given any explicitly bounded[8] and satisfiable system[9] $\mathcal{A}$ of $m$ polynomial constraints in $n$ variables, outputs a level-$\ell$ pseudo-distribution that satisfies $\mathcal{A}$ approximately.*

*Remark* A.7 (Approximation error and bit complexity). For a pseudo-distribution that only approximately satisfies a polynomial system, we can still use sum-of-squares proofs to reason about it in the same way as Lemma A.5. In order for approximation errors not to amplify throughout reasoning, we need to ensure that the bit complexity of the coefficients in the sum-of-squares proof are polynomially bounded.

## A.2 Useful sum-of-squares lemmas

**Lemma A.8.**
$$\{x^2 = x\} \vdash_{2}^{x} 0 \leqslant x \leqslant 1 .$$

*Proof.* The first inequality is trivial due to $\{x^2 = x\} \vdash_{2}^{x} x = x^2 \geqslant 0$. For the second inequality, it follows that
$$\{x^2 = x\} \vdash_{2}^{x} x \leqslant \frac{x^2}{2} + \frac{1}{2} = \frac{x}{2} + \frac{1}{2} .$$

Rearranging the terms, we get
$$\{x^2 = x\} \vdash_{2}^{x} x \leqslant 1 .$$

$\square$

**Lemma A.9.**
$$\{x^2 = x, y^2 = y\} \vdash_{4}^{x,y} 1 - xy \leqslant (1 - x) + (1 - y) .$$

*Proof.* By Lemma A.8, it follows that
$$\{x^2 = x, y^2 = y\} \vdash_{2}^{x,y} 0 \leqslant x, y \leqslant 1 .$$

Therefore, we have
$$\{x^2 = x, y^2 = y\} \vdash_{4}^{x,y} (1 - y)(1 - x) \geqslant 0$$

---

[7]The choice of norm is not important here because the factor $2^{-n^{\ell}}$ swamps the effects of choosing another norm.

[8]A system of polynomial constraints is *explicitly bounded* if it contains a constraint of the form $\|x\|^2 \leqslant M$.

[9]Here we assume that the bit complexity of the constraints in $\mathcal{A}$ is $(n + m)^{O(1)}$.

$$\left|\frac{x,y}{4}\; 1 - x - y \geqslant -xy\right.$$
$$\left|\frac{x,y}{4}\; 2 - x - y \geqslant 1 - xy\right..$$

$\square$

**Lemma A.10.** *Given constant $C$, we have*

$$\{-C \leqslant x \leqslant C\}\left|\frac{x}{2}\; x^2 \leqslant C^2\right..$$

*Proof.*

$$\{-C \leqslant x \leqslant C\}\left|\frac{x}{2}\; (C - x)(C + x) \geqslant 0\right.$$
$$\left|\frac{x}{2}\; C^2 - x^2 \geqslant 0\right.$$
$$\left|\frac{x}{2}\; C^2 \geqslant x^2\right..$$

$\square$

**Lemma A.11.** *Given constant $C$, we have*

$$\{x^2 \leqslant C^2\}\left|\frac{x}{2}\; -C \leqslant x \leqslant C\right..$$

*Proof.* For the first inequality, we have

$$\{x^2 \leqslant C^2\}\left|\frac{x}{2}\; x \geqslant -\frac{x^2}{2C} - \frac{C}{2} \geqslant -\frac{C^2}{2C} - \frac{C}{2} = -C\right..$$

For the second inequality, we have

$$\{x^2 \leqslant C^2\}\left|\frac{x}{2}\; x \leqslant \frac{x^2}{2C} + \frac{C}{2} \leqslant \frac{C^2}{2C} + \frac{C}{2} = C\right..$$

$\square$

## B Concentration inequalities

**Lemma B.1** (Average degree concentration). *Let $Q^\circ$ be an $n$-by-$n$ edge connection probability matrix and let $d^\circ := d(Q^\circ)$. Let $A \sim \mathbb{G}(n, Q^\circ)$. Then for any $\delta \in (0, 1)$,*

$$\mathbb{P}(|d(A) - d^\circ| \geqslant \delta d^\circ) \leqslant 2\exp\left(-\frac{\delta^2 n d^\circ}{6}\right),$$

*Proof.* Let $\mu := \mathbb{E}\sum_{i<j} A_{ij} = \sum_{i<j} p_{ij}$. Using Chernoff bound, for $\delta \in (0, 1)$,

$$\mathbb{P}\left(\left|\sum_{i<j} A_{ij} - \mu\right| \geqslant \delta\mu\right) \leqslant 2\exp\left(-\frac{\delta^2\mu}{3}\right),$$
$$\mathbb{P}(|d(A) - d^\circ| \geqslant \delta d^\circ) \leqslant 2\exp\left(-\frac{\delta^2 n d^\circ}{6}\right).$$

$\square$

**Lemma B.2** (Degree distribution). *Let $Q^\circ$ be an $n$-by-$n$ edge connection probability matrix. Let $d$ be a parameter such that $d \geqslant 5$ and $\|Q^\circ\|_\infty \leqslant d/n$. Then for every $t \in [2e^2, \log n]$, an inhomogeneous random graph $\mathbb{G}(n, Q^\circ)$ has at least $e^{-t}n$ nodes with degree at least $td$ with probability at most $\exp(-te^{-t}nd/4)$.*

*Proof.* Let $m_k$ denote the number of nodes with degree at least $k$ in $\mathbb{G}(n, Q^\circ)$. Then for every $\gamma \in [0, 1]$,

$$\mathbb{P}(m_{td} \geq \gamma n) \leq \binom{n}{\gamma n}\binom{\gamma n^2}{\gamma ntd/2}\left(\frac{d}{n}\right)^{\gamma ntd/2}$$

$$\leq \left(\frac{e}{\gamma}\right)^{\gamma n}\left(\frac{2e}{t}\right)^{\gamma ntd/2}$$

$$= \exp\left(-\gamma n\left(\frac{td}{2}\log\frac{t}{2e} - \log\frac{e}{\gamma}\right)\right).$$

Plugging in $\gamma = e^{-t}$ gives

$$\mathbb{P}(m_{td} \geq e^{-t}n) \leq \exp\left(-te^{-t}n\left(\frac{d}{2}\log\frac{t}{2e} - 1 - 1/t\right)\right).$$

For $t \in [2e^2, \log n]$ and $d \geq 5$,

$$\mathbb{P}(m_{td} \geq e^{-t}n) \leq \exp\left(-te^{-t}n\left(\frac{d}{2} - \frac{5}{4}\right)\right) \leq \exp\left(-te^{-t}nd/4\right).$$

$\square$

**Lemma B.3** (Degree pruning). *Let $Q^\circ$ be an $n$-by-$n$ edge connection probability matrix. Let $d$ be a parameter such that $d \geq 5$ and $\|Q^\circ\|_\infty \leq d/n$. Then with probability at least $1 - n^{1-d/4}$, an inhomogeneous graph $\mathbb{G}(n, Q^\circ)$ has the following property. For all $t \in [2e^2, \log n]$, the number of edges incident to nodes with degree at least $td$ is at most $2te^{-t}nd$;*

*Proof.* Let $m_k$ denote the number of nodes with degree at least $k$ in $\mathbb{G}(n, Q^\circ)$. By Lemma B.2, for any $t \in [2e^2, \log n]$ and $d \geq 5$,

$$\mathbb{P}(m_{td} \geq e^{-t}n) \leq \exp\left(-te^{-t}nd/4\right) \leq n^{-d/4}.$$

Applying union bound, the event that $m_k \leq e^{-k/d}n$ for any integer $k \in [2e^2d, (\log n)d]$ happens with probability at least $1 - n^{1-d/4}$. We condition our following analysis on this event.

Fix a $t \in [2e^2, \log n]$. The number of edges incident to nodes with degree at least $td$ is at most

$$\sum_{i=0}(t+i+1)d \cdot e^{-(t+i)}n = nd\sum_{i=t}(i+1)e^{-i} = nde^{-t}\left(\frac{e}{e-1}t + \frac{e^2}{(e-1)^2}\right) \leq 2te^{-t}nd.$$

$\square$

**Lemma B.4** (Degree-truncated subgraph). *Let $Q^\circ$ be an $n$-by-$n$ edge connection probability matrix and let $d^\circ := d(Q^\circ)$. Let $d$ be a parameter such that $d \geq 5$ and $\|Q^\circ\|_\infty \leq d/n$. For $\delta \in (0, 1)$, an inhomogeneous graph $A \sim \mathbb{G}(n, Q^\circ)$ has the following property with probability at least $1 - n^{1-d/4} - \exp(-\delta^2 nd^\circ/6)$. For every $t \in [2e^2, \log n]$, $A$ contains an $n$-node subgraph $\tilde{A}$ of such that*

- $(\tilde{A}\mathbb{1})_i \leq td$ *for any $i \in [n]$;*

- $(1 - \delta)d^\circ - 4te^{-t}d \leq d(\tilde{A}) \leq (1 + \delta)d^\circ.$

*Proof.* By Lemma B.1 and Lemma B.3, $A \sim \mathbb{G}(n, Q^\circ)$ has the following two properties with probability at least $1 - n^{1-d/4} - \exp(-\delta^2 nd^\circ/6)$.

- $|d(A) - d^\circ| \leq \delta d^\circ.$

- For all $t \in [2e^2, \log n]$, the number of edges incident to nodes with degree at least $td$ is at most $2te^{-t}nd$.

Consider a graph $A$ with the above two properties. Fix a $t \in [2e^2, \log n]$. By removing at most $2te^{-t}nd$ edges from $A$, we can obtain a graph $\tilde{A}$ such that the maximum degree of $\tilde{A}$ is at most $td$. Moreover,

$$d(\tilde{A}) \geqslant d(A) - 4te^{-t}d \geqslant (1 - \delta)d^\circ - 4te^{-t}d \,,$$
$$d(\tilde{A}) \leqslant d(A) \leqslant (1 + \delta)d^\circ \,.$$

$\square$

**Lemma B.5** (Spectral bound [BvH16]). *Let $A \sim \mathbb{G}(n, p_0)$ and suppose $np_0 \geqslant 5$. Then with probability at least $1 - n^{-\Omega(1)}$,*

$$\left\| A - p_0(\mathbb{1}\mathbb{1}^\top - \mathrm{Id}) \right\|_{\mathrm{op}} \leqslant O\left( \sqrt{np_0 \log n} \right).$$

## C  Sum-of-squares exponential mechanism

In this section, we present our sum-of-squares exponential mechanism and prove its properties in a general setting that incorporates all special cases in Appendix D, Appendix E and Appendix F.

**Setup.**  Let $\mathcal{D} \subset \mathbb{R}^N$. Given an $n$-by-$n$ symmetric matrix $A$, our goal is to output an element $d$ from $\mathcal{D}$ privately. We say two symmetric matrices are neighboring if they differ in at most one row and one column. The utility of an element $d \in \mathcal{D}$ is quantified by a score function defined as follows.

**Score function.**  For an $n$-by-$n$ symmetric matrix $A$ and a scalar $\gamma$, consider the following polynomial system with indeterminates $(Y_{ij})_{i,j \in [n]}$, $(z_i)_{i \in [n]}$ and coefficients that depend on $A, \gamma$:

$$Q_1(Y, z; A, \gamma) := \begin{cases} z \odot z = z, \ \langle \mathbb{1}, z \rangle \geqslant (1 - \gamma)n \\ 0 \leqslant Y \leqslant \mathbb{1}\mathbb{1}^\top, \ Y = Y^\top \\ Y \odot zz^\top = A \odot zz^\top \end{cases}. \tag{C.1}$$

For an element $d \in \mathcal{D}$, let $Q_2(Y; d)$ be a polynomial system with coefficients depending on $d$ (and independent of $A, \gamma$). Then for a matrix $A$ and an element $d \in \mathcal{D}$, we define the score of $d$ with regard to $A$ to be

$$s(d; A) := \min_{0 \leqslant \gamma \leqslant 1} \gamma n \ \text{s.t.} \begin{cases} \exists \ \text{level-}\ell \ \text{pseudo-expectation} \ \tilde{\mathbb{E}} \ \text{satisfying} \\ Q_1(Y, z; A, \gamma) \ \cup \ Q_2(Y; d) \end{cases}. \tag{C.2}$$

For $s(d; A)$ to be well-defined, we assume that for every $d \in \mathcal{D}$ there exists a symmetric matrix $A^* \in [0, 1]^{n \times n}$ such that $Q_2(A^*; d)$ is true.

*Remark* C.1 (Score function computation).  Observe that a level-$\ell$ pseudo-expectation satisfying $Q_1(Y, z; A, \gamma) \cup Q_2(Y; d)$ is also a level-$\ell$ pseudo-expectation satisfying $Q_1(Y, z; A, \gamma') \cup Q_2(Y; d)$ for any $\gamma' \geqslant \gamma$. Thus we can compute $s(d; A)$ using binary search. Given a scalar $\gamma$, checking if there exists a level-$\ell$ pseudo-expectation satisfying $Q_1(Y, z; A, \gamma) \cup Q_2(Y; d)$ is equivalent to checking if a semidefinite program of size $n^{O(\ell)}$ is feasible. Since we only have efficient algorithms for semidefinite programming up to a given precision, we can only efficiently search for pseudo-distributions that *approximately* satisfy a given polynomial system. In spite of this, as long as the bit complexity of the coefficients in our sum-of-squares proof are polynomially bounded, the analysis of our algorithm based on sum-of-squares proofs will still work due to our discussion in Remark A.7. We refer interested readers to [HKMN23] for a formal (and quite technical) treatment of approximate pseudo-expectations.

**Exponential mechanism.**  Given an $n$-by-$n$ symmetric matrix $A$, our sos exponential mechanism with privacy parameter $\varepsilon$ outputs a sample from the distribution $\mu_{A,\varepsilon}$ that is supported on $\mathcal{D}$ and has density

$$\mathrm{d}\mu_{A,\varepsilon}(d) \propto \exp(-\varepsilon \cdot s(d; A)). \tag{C.3}$$

*Remark* C.2 (Sampling). To efficiently sample from $\mu_{A,\varepsilon}$, we can use the following straightforward discretization scheme. More specifically, given a discretization parameter $\delta$, we output an element $d \in \{0, \delta, 2\delta, \ldots, \lfloor n/\delta \rfloor \delta\}$ with probability proportional to $\exp(-\varepsilon \cdot \text{sos-score}(d; A))$. As the error introduced by discretization is at most $\delta$ and our target estimation error is $\omega(1/n)$, we can choose $\delta = 1/n$ and the discretization error is then negligible. Moreover, our algorithm requires at most $n^2$ evaluations of score functions.

**Properties.** The following lemma shows that the sensitivity of score function $s(d; A)$ is at most 1.

**Lemma C.3** (Sensitivity bound). *For any $d \in \mathcal{D}$ and any two n-by-n symmetric matrices $A, A'$ that differ in at most one row and one column, the score function defined in Eq. (C.2) satisfies*

$$|s(d; A) - s(d; A')| \leqslant 1 \,.$$

*Proof.* Without loss of generality, we assume that $A$ and $A'$ differ in the first row and column. Consider the linear functions $(\ell_i)$ where $\ell_1(z) = 0$ and $\ell_i(z) = z_i$ for $i \geqslant 2$. Then for every polynomial inequality $q(Y, z) \geqslant 0$ in $Q_1(Y, z; A', \gamma + 1/n) \cup Q_2(Y; d)$,

$$Q_1(Y, z; A, \gamma) \cup Q_2(Y; d) \,\Big|_{\overline{\deg(q)}}^{Y,z} \; q(Y, \ell(z)) \geqslant 0 \,.$$

The same argument also holds for polynomial equalities. Then by [CDd+24, Lemma 8.1], $s(d; A') \leqslant s(d; A) + 1$. Due to symmetry of $A$ and $A'$, we also have $s(d; A) \leqslant s(d; A') + 1$. Therefore, $|s(d; A) - s(d; A')| \leqslant 1$. $\qquad\square$

The following privacy guarantee of our sos exponential mechanism is a direct corollary of Lemma C.3.

**Lemma C.4** (Privacy). *The exponential mechanism defined in Eq. (C.3) is $2\varepsilon$-differentially node private.*

**Lemma C.5** (Volume of low-score points). *Let $A \in \mathbb{R}^{n \times n}$ and $\varepsilon > 0$. Consider the distribution $\mu_{A,\varepsilon}$ defined by Eq. (C.3). Suppose $(Y = A^*, z = z^*)$ is a solution to $Q_1(Y, z; A, \gamma^*)$. Then for any $t \geqslant 0$,*

$$\mathbb{P}_{d \sim \mu_{A,\varepsilon}} \left( s(d; A) \geqslant \gamma^* n + \frac{t \log n}{\varepsilon} \right) \leqslant \frac{\text{vol}(\mathcal{D})}{\text{vol}(\mathcal{G}(A^*))} \cdot n^{-t} \,,$$

*where $\mathcal{G}(A^*) := \{d \in \mathcal{D} \,:\, Q_2(A^*; d) \text{ is true}\}$.*

*Proof.* Note $(Y = A^*, z = z^*)$ is also a solution to $Q_1(Y, z; A, \gamma^*) \cup Q_2(Y; d)$ for any $d$ such that $Q_2(A^*; d)$ is true. Let $\mathcal{G}(A^*) := \{d \in \mathcal{D} \,:\, Q_2(A^*; d) \text{ is true}\}$. Thus $s(d; A) \leqslant \gamma^* n$ for any $d \in \mathcal{G}(A^*)$. For $t \geqslant 0$,

$$\mathbb{P}_{d \sim \mu_{A,\varepsilon}} \left( s(d; A) \geqslant \gamma^* n + \frac{t \log n}{\varepsilon} \right) \leqslant \frac{\text{vol}(\mathcal{D}) \cdot \exp\left(-\varepsilon \gamma^* n - t \log n\right)}{\text{vol}(\mathcal{G}(A^*)) \cdot \exp(-\varepsilon \gamma^* n)} = \frac{\text{vol}(\mathcal{D})}{\text{vol}(\mathcal{G}(A^*))} \cdot n^{-t} \,.$$

$\qquad\square$

# D   Coarse estimation

In this section, we describe our coarse estimation algorithm that achieves constant multiplicative approximation of the expected average degree $d^\circ$.

**Theorem D.1** (Coarse estimation for inhomogeneous random graphs). *Let $Q^\circ$ be an n-by-n edge connection probability matrix and let $d^\circ := d(Q^\circ)$. Suppose $\|Q^\circ\|_\infty \leqslant R d^\circ / n$ for some $R$. There are constants $C_1, C_2, C_3$ such that the following holds. For any $\eta, \varepsilon, d^\circ$ such that $\eta \log(1/\eta) R \leqslant C_1$, $\varepsilon \geqslant C_2 \log^2(n) R / n$, and $d^\circ \geqslant C_3$, there exists a polynomial-time $\varepsilon$-differentially node private algorithm which, given an $\eta$-corrupted inhomogeneous random graph $\mathbb{G}(n, Q^\circ)$, outputs an estimate $\hat{d}$ satisfying $|\hat{d}/d^\circ - 1| \leqslant 0.5$ with probability $1 - n^{-\Omega(1)}$.*

We make a few remarks on Theorem D.1.

- Our algorithm in Theorem D.1 is a sum-of-squares exponential mechanism. $R, \eta, \varepsilon$ are parameters given as inputs to our algorithm.

- We can get a constant estimate of $p^\circ$ by taking $\hat{p} = \frac{\hat{d}}{n-1}$. Since $\frac{\hat{p}}{p^\circ} = \frac{\hat{d}}{d^\circ}$, it follows that $|\frac{\hat{p}}{p^\circ} - 1| \leq 0.5$.

- When $Q^\circ = p^\circ(\mathbb{1}\mathbb{1}^\top - \mathrm{Id})$, the inhomogeneous random graph $\mathbb{G}(n, Q^\circ)$ is just the Erdős-Rényi random graph $\mathbb{G}(n, p^\circ)$. Thus, by setting $R = \frac{n}{n-1}$ in Theorem D.1, we directly obtain a coarse estimation result for Erdős-Rényi random graphs.

- The utility guarantee of our algorithm holds in the constant-degree regime (i.e. $d^\circ \geq \Omega(1)$). To the best of our knowledge, even without privacy requirement and in the special case of Erdős-Rényi random graphs, no previous algorithm can match our guarantees in the constant-degree regime. Specifically, when $d^\circ \ll \log n$ and $\eta \geq \Omega(1)$, the robust algorithm in [AJK$^+$22] can not provide a constant-factor approximation of $d^\circ$.

In Appendix D.1, we set up polynomial systems that our algorithm uses and prove useful sos inequalities. In Appendix D.2, we show that we can easily obtain a robust algorithm via sos proofs in Appendix D.1. Then in Appendix D.3, we describe our algorithm and prove Theorem D.1.

## D.1 Sum-of-squares

For an adjacency matrix $A$ and two nonnegative scalars $\gamma$ and $\sigma$, consider the following polynomial systems with indeterminates $Y = (Y_{ij})_{i,j\in[n]}$, $z = (z_i)_{i\in[n]}$ and coefficients that depend on $A, \gamma, \sigma$:

$$\mathcal{P}_1(Y, z; A, \gamma) := \begin{cases} z \odot z = z, \ \langle \mathbb{1}, z \rangle \geq (1 - \gamma)n \\ 0 \leq Y \leq \mathbb{1}\mathbb{1}^\top, \ Y = Y^\top \\ Y \odot zz^\top = A \odot zz^\top \end{cases}, \tag{D.1}$$

$$\mathcal{P}_2(Y; \sigma) := \begin{cases} d(Y) = \langle Y, \mathbb{1}\mathbb{1}^\top \rangle / n \\ (Y\mathbb{1})_i \leq \sigma d(Y) \qquad \forall i \in [n] \end{cases}. \tag{D.2}$$

For convenience of notation, we will consider the following combined polynomial system in remaining of the section

$$C(Y, z; A, \gamma, \sigma) := \mathcal{P}_1(Y, z; A, \gamma) \cup \mathcal{P}_2(Y; \sigma). \tag{D.3}$$

**Lemma D.2.** *If $(A^*, z^*)$ is a feasible solution to $C(Y, z; A, \gamma^*, \sigma)$ and $1 - 2\gamma\sigma - 2\gamma^*\sigma > 0$, then it follows that*

$$C(Y, z; A, \gamma, \sigma) \Big|_{8}^{Y, z} (1 - 2\gamma\sigma - 2\gamma^*\sigma)d(A^*) \leq d(Y) \leq \frac{1}{1 - 2\gamma\sigma - 2\gamma^*\sigma}d(A^*).$$

*Proof.* Let $w = z \odot z^*$, by constraint $Y \odot zz^\top = A \odot zz^\top$ and $A^* \odot z^*(z^*)^\top = A \odot z^*(z^*)^\top$, we have

$$C \Big|_{4}^{Y,z} Y \odot ww^\top = Y \odot zz^\top \odot z^*(z^*)^\top$$
$$= A \odot zz^\top \odot z^*(z^*)^\top$$
$$= A \odot z^*(z^*)^\top \odot zz^\top$$
$$= A^* \odot z^*(z^*)^\top \odot zz^\top$$
$$= A^* \odot ww^\top.$$

Applying this equality, it follows that

$$C \Big|_{4}^{Y,z} n \cdot d(Y) = \langle Y, \mathbb{1}\mathbb{1}^\top \rangle$$
$$= \langle Y, ww^\top \rangle + \langle Y, \mathbb{1}\mathbb{1}^\top - ww^\top \rangle$$

$$= \left\langle A^*, ww^\top \right\rangle + \left\langle Y, 2(\mathbb{1} - w)\mathbb{1}^\top \right\rangle - \left\langle Y, (\mathbb{1} - w)(\mathbb{1} - w)^\top \right\rangle.$$

For the first term, since $A^*_{i,j} \in [0,1]$, $z^*_i \in \{0,1\}$ and $C \vdash^{Y,z}_{2} 0 \leqslant z_i \leqslant 1$ for all $i, j \in [n]$, we have

$$C \vdash^{Y,z}_{4} A^*_{i,j} w_i w_j = A^*_{i,j} z^*_i z^*_j z_i z_j \leqslant A^*_{i,j}.$$

Therefore, it follows that

$$C \vdash^{Y,z}_{4} \left\langle A^*, ww^\top \right\rangle \leqslant \left\langle A^*, \mathbb{1}\mathbb{1}^\top \right\rangle \leqslant n \cdot d(A^*). \tag{D.4}$$

For the second term, we have

$$
\begin{aligned}
C \vdash^{Y,z}_{4} \left\langle Y, 2(\mathbb{1} - w)\mathbb{1}^\top \right\rangle &= \langle Y\mathbb{1}, 2(\mathbb{1} - w) \rangle \\
&= \sum_{i \in [n]} 2(1 - w_i) \cdot (Y\mathbb{1})_i \\
&= \sum_{i \in [n]} 2(1 - z_i z^*_i) \cdot (Y\mathbb{1})_i \\
&\leqslant \sum_{i \in [n]} 2(1 - z_i) \cdot (Y\mathbb{1})_i + \sum_{i \in [n]} 2(1 - z^*_i) \cdot (Y\mathbb{1})_i,
\end{aligned}
$$

where the last inequality is due to Lemma A.9. From constraints $\sum_{i \in [n]} 1 - z^*_i \leqslant \gamma^* n$, $\sum_{i \in [n]} 1 - z_i \leqslant \gamma n$ and $(Y\mathbb{1})_i \leqslant \sigma d(Y)$ for all $i \in [n]$, it follows that

$$
\begin{aligned}
C \vdash^{Y,z}_{4} \left\langle Y, 2(\mathbb{1} - w)\mathbb{1}^\top \right\rangle &\leqslant \sum_{i \in [n]} 2(1 - z_i) \cdot \sigma d(Y) + \sum_{i \in [n]} 2(1 - z^*_i) \cdot \sigma d(Y) \\
&= 2\sigma d(Y) \cdot \left( \sum_{i \in [n]} 1 - z_i \right) + 2\sigma d(Y) \cdot \left( \sum_{i \in [n]} 1 - z^*_i \right) \\
&\leqslant 2\gamma n \sigma d(Y) + 2\gamma^* n \sigma d(Y).
\end{aligned}
\tag{D.5}
$$

For the third term, since $C \vdash^{Y,z}_{2} Y_{i,j} \geqslant 0$ and $C \vdash^{Y,z}_{2} 1 - w_i \geqslant 0$ for all $i, j \in [n]$, it follows that

$$C \vdash^{Y,z}_{8} \left\langle Y, (\mathbb{1} - w)(\mathbb{1} - w)^\top \right\rangle \geqslant 0. \tag{D.6}$$

Combining Eq. (D.4), Eq. (D.5) and Eq. (D.6), we can get

$$C \vdash^{Y,z}_{8} n \cdot d(Y) \leqslant n \cdot d(A^*) + 2\gamma n \sigma d(Y) + 2\gamma^* n \sigma d(Y)$$

$$\vdash^{Y,z}_{8} d(Y) \leqslant \frac{d(A^*)}{1 - 2\gamma\sigma - 2\gamma^*\sigma}.$$

Swapping the roll of $A^*$ and $Y$, we can use the same proof to get

$$C \vdash^{Y,z}_{8} n \cdot d(A^*) \leqslant n \cdot d(Y) + 2\gamma n \sigma d(A^*) + 2\gamma^* n \sigma d(A^*)$$

$$\vdash^{Y,z}_{8} (1 - 2\gamma\sigma - 2\gamma^*\sigma)d(A^*) \leqslant d(Y).$$

This completes the proof. $\qquad \square$

**Lemma D.3.** *Let $Q^\circ$ be an n-by-n edge connection probability matrix and $d^\circ := d(Q^\circ)$. Suppose $\|Q^\circ\|_\infty \leqslant R d^\circ/n$ for $R \in \mathbb{R}$. Let $A$ be an $\eta$-corrupted adjacency matrix of a random graph $G^\circ \sim \mathbb{G}(n, Q^\circ)$. Suppose $\eta \log(1/\eta)R \leqslant C_1$ for some constant $C_1$ that is small enough. With probability $1 - n^{-\Omega(1)}$, there exists $A^*$ and $z^*$ such that*

1. *$|d(A^*) - d^\circ| \leqslant 0.1 d^\circ$.*

2. *$(A^*, z^*)$ is a feasible solution to $C(Y, z; A, \gamma, \sigma)$ with $\gamma = 2\eta$ and $\sigma = 2\log(1/\eta)R$.*

*Proof.* Let $A^\circ$ be the adjacency matrix of $G^\circ$ and $z^\circ \in \{0,1\}^n$ denote the set of uncorrupted nodes ($z_i^\circ = 1$ if and only if node $i$ is uncorrupted).

By Lemma B.2 and Lemma B.3, we know that, with probability $1 - n^{-\Omega(1)}$, there exists a degree-pruned adjacency matrix $\tilde{A}$ such that

1. $\left\|\tilde{A}\mathbb{1}\right\|_\infty \leqslant \log(1/\eta)Rd^\circ$.

2. At most $\eta n$ nodes are pruned.

3. At most $2\eta \log(1/\eta)nRd^\circ$ edges are pruned.

Let $\tilde{z} \in \{0,1\}^n$ denote the set of unpruned nodes ($z_i^\circ = 1$ if and only if node $i$ is not pruned). We will show that $A^* = \tilde{A}$ and $z^* = z^\circ \odot \tilde{z}$ satisfies the lemma.

**Guarantee 1.** By Lemma B.1, we know that, with probability $1 - n^{-\Omega(1)}$,

$$|d(A^\circ) - d^\circ| \leqslant 10\sqrt{\frac{d^\circ \log n}{n}} . \tag{D.7}$$

From degree pruning guarantee (3), we have that

$$|d(\tilde{A}) - d(A^\circ)| \leqslant 2\eta \log(1/\eta)nRd^\circ . \tag{D.8}$$

Combining Eq. (D.7) and Eq. (D.8), for some constant $C_1$ that is small enough, we have

$$
\begin{aligned}
|d(\tilde{A}) - d^\circ| &\leqslant |d(\tilde{A}) - d(A^\circ)| + |d(A^\circ) - d^\circ| \\
&\leqslant 10\sqrt{\frac{d^\circ \log n}{n}} + 2\eta \log\frac{1}{\eta}Rd^\circ \\
&\leqslant 10\sqrt{\frac{\log n}{n}}d^\circ + 2C_1 d^\circ \\
&\leqslant 0.1d^\circ .
\end{aligned}
\tag{D.9}
$$

**Guarantee 2.** It is easy to check that $z^* \odot z^* = z^*$, $0 \leqslant A^* \leqslant \mathbb{1}\mathbb{1}^\top$ and $A^* = (A^*)^\top$. Since $\langle \mathbb{1}, \tilde{z} \rangle \geqslant 1 - \eta n$ by degree pruning condition (2) and $\langle \mathbb{1}, z^\circ \rangle \geqslant 1 - \eta n$ by corruption rate, it is easy to verify that
$$\langle \mathbb{1}, z^* \rangle \geqslant 1 - 2\eta n .$$
Moreover, we have $A^* \odot z^*(z^*)^\top = A \odot z^*(z^*)^\top$ due to

$$\tilde{A} \odot \tilde{z}\tilde{z}^\top \odot z^\circ(z^\circ)^\top = A^\circ \odot \tilde{z}\tilde{z}^\top \odot z^\circ(z^\circ)^\top = A^\circ \odot z^\circ(z^\circ)^\top \odot \tilde{z}\tilde{z}^\top = A \odot z^\circ(z^\circ)^\top \odot \tilde{z}\tilde{z}^\top .$$

From Eq. (D.9), we can get that $d^\circ \leqslant 2d(\tilde{A})$. Plugging this into degree pruning condition (1), we get

$$\left\|\tilde{A}\mathbb{1}\right\|_\infty \leqslant \log(1/\eta)Rd^\circ \leqslant 2\log(1/\eta)Rd(\tilde{A}) .$$

Therefore, we have

$$(A^*\mathbb{1})_i \leqslant 2\log(1/\eta)Rd(A^*) .$$

for all $i \in [n]$.

Thus, $(A^*, z^*)$ is a feasible solution to $C(Y, z; A, \gamma, \sigma)$ with $\gamma = 2\eta$ and $\sigma = 2\log(1/\eta)R$. $\qquad \square$

## D.2 Robust algorithm

In this section, we show that the following algorithm based on sum-of-squares proofs in Appendix D.1 obtains a robust constant multiplicative approximation of $d^\circ$.

> **Algorithm D.4** (Robust coarse estimation algorithm).
> **Input:** $\eta$-corrupted adjacency matrix $A$, corruption fraction $\eta$ and parameter $R$.
>
> **Algorithm:** Obtain level-8 pseudo-expectation $\tilde{\mathbb{E}}$ by solving sum-of-squares relaxation of program $C(Y, z; A, \gamma, \sigma)$ (defined in Eq. (D.3)) with $A$, $\gamma = 2\eta$ and $\sigma = 2\log(1/\eta)R$.
>
> **Output:** $\tilde{\mathbb{E}}[d(Y)]$

**Theorem D.5** (Robust coarse estimation). *Let $Q^\circ$ be an n-by-n edge connection probability matrix and let $d^\circ := d(Q^\circ)$. Suppose $\|Q^\circ\|_\infty \leqslant Rd^\circ/n$ for some $R$. Let $A$ be an $\eta$-corrupted adjacency matrix of a random graph $G^\circ \sim \mathbb{G}(n, Q^\circ)$. Suppose $\eta\log(1/\eta)R \leqslant c$ for some constant $c$ that is small enough. With probability $1 - n^{-\Omega(1)}$, Algorithm D.4 outputs an estimate $\hat{d}$ satisfying $|\frac{\hat{d}}{d^\circ} - 1| \leqslant 0.5$.*

*Proof.* By Lemma D.2 and Lemma D.3, we know that

$$C(Y, z; A, \gamma, \sigma)\Big|_{8}^{Y,z} (1 - 4\gamma\sigma)d(A^*) \leqslant d(Y) \leqslant \frac{1}{1 - 4\gamma\sigma}d(A^*),$$

and,

$$|d(A^*) - d^\circ| \leqslant 0.1d^\circ.$$

Therefore, we have

$$C(Y, z; A, \gamma, \sigma)\Big|_{8}^{Y,z} 0.9(1 - 4\gamma\sigma)d^\circ \leqslant d(Y) \leqslant \frac{1.1}{1 - 4\gamma\sigma}d^\circ.$$

Consider $4\gamma\sigma$, for constant $c$ that is small enough, we have

$$4\gamma\sigma = 8\eta\log(1/\eta)R \leqslant 8c \leqslant 0.1.$$

This implies that $0.9(1 - 4\gamma\sigma) \geqslant \frac{1}{2}$ and $\frac{1.1}{1-4\gamma\sigma} \leqslant \frac{11}{9} \leqslant \frac{3}{2}$. Therefore, we have

$$C(Y, z; A, \gamma, \sigma)\Big|_{8}^{Y,z} \frac{1}{2}d^\circ \leqslant d(Y) \leqslant \frac{3}{2}d^\circ.$$

Thus, the level-8 pseudo-expectation $\tilde{\mathbb{E}}$ satisfies

$$\frac{1}{2}d^\circ \leqslant \tilde{\mathbb{E}}[d(Y)] \leqslant \frac{3}{2}d^\circ,$$

which implies that

$$\left|\frac{\tilde{\mathbb{E}}[d(Y)]}{d^\circ} - 1\right| \leqslant \frac{1}{2}.$$

$\square$

## D.3 Private algorithm

In this section, we present our algorithm and prove Theorem D.1. Our algorithm instantiates the sum-of-squares exponential mechanism in Appendix C.

**Score function.** For an $n$-by-$n$ symmetric matrix $A$ and a scalar $d$, we define the score of $d$ with regard to $A$ to be

$$s(d; A) := \min_{0 \leqslant \gamma \leqslant 1} \gamma n \text{ s.t. } \begin{cases} \exists \text{ level-8 pseudo-expectation } \tilde{\mathbb{E}} \text{ satisfying} \\ C(Y, z; A, \gamma, \sigma) \cup \{|d(Y) - d| \leqslant \alpha d\}, \end{cases} \tag{D.10}$$

where $C(Y, z; A, \gamma, \sigma)$ is the polynomial system defined in Eq. (D.3), and $\sigma, \alpha$ are fixed parameters whose values will be decided later. Note that $(Y = \frac{d}{n}\mathbb{1}\mathbb{1}^\top, z = \mathbb{0})$ is a solution to the polynomial system $C(Y, z; A, 1, \sigma) \cup \{|d(Y) - d| \leqslant \alpha d\}$ for any $A \in \mathbb{R}^{n \times n}, d \in [0, n]$, and $\sigma \geqslant 1$.

To efficiently compute $s(d; A)$, we can use the scheme as described in Remark C.1.

**Exponential mechanism.** Given a privacy parameter $\varepsilon$ and an $n$-by-$n$ symmetric matrix $A$, our algorithm is the exponential mechanism with score function Eq. (D.10) and range $[0, n]$.

---

**Algorithm D.6** (Coarse estimation).
**Input:** Graph $A$.

**Parameters:** $\varepsilon, \sigma, \alpha$.

**Output:** A sample from the distribution $\mu_{A,\varepsilon}$ with support $[0, n]$ and density

$$\mathrm{d}\mu_{A,\varepsilon}(d) \propto \exp(-\varepsilon \cdot s(d; A)), \tag{D.11}$$

where $s(d; A)$ is defined in Eq. (D.10).

---

To efficiently sample from $\mu_{A,\varepsilon}$, we can use the scheme as described in Remark C.2.

**Privacy.** The following privacy guarantee of our algorithm is a direct corollary of Lemma C.4.

**Lemma D.7** (Privacy). *Algorithm D.6 is $2\varepsilon$-differentially node private.*

**Utility.** The utility guarantee of our algorithm is stated in the following lemma.

**Lemma D.8** (Utility). *Let $Q^\circ$ be an n-by-n edge connection probability matrix and let $d^\circ := d(Q^\circ)$. Suppose $\|Q^\circ\|_\infty \leqslant Rd^\circ/n$ for some $R$. There are constants $C_1, C_2, C_3$ such that the following holds. For any $\eta, \varepsilon, d^\circ$ such that $\eta \log(1/\eta)R \leqslant C_1$, $\varepsilon \geqslant C_2 \log^2(n)R/n$, and $d^\circ \geqslant C_3$, given an $\eta$-corrupted inhomogeneous random graph $\mathbf{G}(n, Q^\circ)$, Algorithm D.6 outputs an estimate $\hat{d}$ satisfying $|\hat{d} - d^\circ| \leqslant 0.5d^\circ$ with probability $1 - n^{-\Omega(1)}$.*

Before proving Lemma D.8, we need the following two lemmas.

**Lemma D.9** (Volume of low-score points). *Let $A \in \mathbb{R}^{n \times n}$ and $\varepsilon > 0$. Consider the distribution $\mu_{A,\varepsilon}$ defined by Eq. (D.11). Suppose $(Y = A^*, z = z^*)$ is a solution to $C(Y, z; A, \gamma^*, \sigma)$ and $d(A^*) \geqslant 2$. Then for any $t \geqslant 0$,*

$$\mathbb{P}_{d \sim \mu_{A,\varepsilon}}\left(s(d; A) \geqslant \gamma^* n + \frac{t \log n}{\varepsilon}\right) \leqslant \frac{n^{-t+1}}{\alpha}.$$

*Proof.* Apply Lemma C.5 with $\mathcal{D} = [0, n]$ and

$$\mathcal{G}(A^*) = \left\{d \in \mathcal{D} : \frac{d(A^*)}{1 + \alpha} \leqslant d \leqslant \frac{d(A^*)}{1 - \alpha}\right\}.$$

As $[d(A^*)/(1 + \alpha), d(A^*)] \subseteq \mathcal{G}(A^*)$ and $d(A^*) \geqslant 2 \geqslant 1 + \alpha$, we have $\mathrm{vol}(\mathcal{G}(A^*)) \geqslant \alpha$. $\square$

**Lemma D.10** (Low score implies utility). *Let $A \in \mathbb{R}^{n \times n}$ and consider the score function $s(\cdot\, ; A)$ defined in Eq. (D.10). Suppose $(Y = A^*, z = z^*)$ is a solution to $C(Y, z; A, \gamma^*, \sigma)$. For a scalar $d$ such that $s(d; A) \leqslant \tau n$ and $(\gamma^* + \tau)\sigma \leqslant 0.1$,*

$$\frac{0.8}{1 + \alpha}d(A^*) \leqslant d \leqslant \frac{1.25}{1 - \alpha}d(A^*).$$

*Proof.* Applying Lemma D.2 with $(\gamma^* + \tau)\sigma \leqslant 0.1$, we have

$$C(Y, z; A, \tau, \sigma) \;\Big|_{8}^{Y,z}\; 0.8d(A^*) \leqslant d(Y) \leqslant 1.25d(A^*)$$

Thus,

$$C(Y, z; A, \tau, \sigma) \cup \{|d(Y) - d| \leqslant \alpha d\} \;\Big|_{8}^{Y,z}\; \frac{0.8}{1 + \alpha}d(A^*) \leqslant d \leqslant \frac{1.25}{1 - \alpha}d(A^*).$$

$\square$

Now we are ready to prove Lemma D.8.

*Proof of Lemma D.8.* Let $A$ be a realization of $\eta$-corrupted $\mathbb{G}(n, Q^\circ)$. By Lemma D.3, the following event happens with probability $1 - n^{-\Omega(1)}$. There exists a solution $(Y = A^*, z = z^*)$ to $C(Y, z; A, \gamma^*, \sigma)$ with $\gamma^* = 2\eta$, $\sigma = 2\log(1/\eta)R$, and $0.9d^\circ \leqslant d(A^*) \leqslant 1.1d^\circ$.

As $d(A^*) \geqslant 0.9d^\circ \geqslant 2$, then it follows by setting $t = 10$ and $\alpha = 0.01$ in Lemma D.9 that,

$$\mathop{\mathbb{P}}_{d \sim \mu_{A,\varepsilon}} (s(\boldsymbol{d}; A) \leqslant \tau n) \geqslant 1 - n^{-9} \text{ where } \tau := 2\eta + 10\log(n)/(\varepsilon n).$$

As an $\eta$-corrupted graph is actually uncorrupted when $\eta < 1/n$, we can assume $\eta \geqslant 1/(2n)$ without loss of generality. Thus,

$$(2\eta + \tau)\sigma \leqslant 8\eta \log(1/\eta)R + \frac{20\log^2(n)R}{n\varepsilon}.$$

For $\eta \log(1/\eta)R$ and $\log^2(n)R/(\varepsilon n)$ smaller than some constant, we have $(2\eta + \tau)\sigma \leqslant 0.1$. Let $\hat{d}$ be a scalar such that $s(\hat{d}; A) \leqslant \tau n$. Then by Lemma D.10,

$$\frac{0.8}{1 + \alpha}d(A^*) \leqslant \hat{d} \leqslant \frac{1.25}{1 - \alpha}d(A^*).$$

Plugging in $\alpha \leqslant 0.01$ and $0.9d^\circ \leqslant d(A^*) \leqslant 1.1d^\circ$, we have

$$0.5d^\circ \leqslant \hat{d} \leqslant 1.5d^\circ.$$

$\square$

**Proof of Theorem D.1.** By Lemma D.7 and Lemma D.8.

# E  Fine estimation for inhomogeneous random graphs

From Appendix D, we have a constant multiplicative approximation of the expected average degree $d^\circ$. In this section, we show how to use this coarse estimate to obtain our fine estimator for inhomogeneous random graphs.

**Theorem E.1** (Fine estimation for inhomogeneous random graphs)**.** *Let $Q^\circ$ be an $n$-by-$n$ edge connection probability matrix and let $d^\circ := \bar{d}(Q^\circ)$. Suppose $\|Q^\circ\|_\infty \leqslant Rd^\circ/n$ for some $R$. There is a sufficiently small constant $c$ such that the following holds. For any $\eta$ such that $\eta \log(1/\eta)R \leqslant c$, there exists a polynomial-time $\varepsilon$-differentially node private algorithm which, given an $\eta$-corrupted inhomogeneous random graph $\mathbb{G}(n, Q^\circ)$ and a constant-factor approximation of $d^\circ$, outputs an estimate $\tilde{d}$ satisfying*

$$\left|\frac{\tilde{d}}{d^\circ} - 1\right| \leqslant O\left(\sqrt{\frac{\log n}{d^\circ n}} + \frac{R\log^2 n}{\varepsilon n} + \eta \log(1/\eta)R\right),$$

*with probability $1 - n^{-\Omega(1)}$.*

We make a few remarks on Theorem E.1.

- Our algorithm in Theorem E.1 is a sum-of-squares exponential mechanism. $R, \eta, \varepsilon$ are parameters given as input to our algorithm.

- We can get an estimate of $p^\circ$ by taking $\tilde{p} = \frac{\tilde{d}}{n-1}$. Since $\frac{\tilde{p}}{p^\circ} = \frac{\tilde{d}}{d^\circ}$, it follows that

$$\left|\frac{\tilde{p}}{p^\circ} - 1\right| \leqslant O\left(\sqrt{\frac{\log n}{d^\circ n}} + \frac{R\log^2 n}{\varepsilon n} + \eta \log(1/\eta)R\right).$$

- Combining Theorem D.1 and Theorem E.1 gives us an efficient, private, and robust edge density estimation algorithm for inhomogeneous random graphs whose utility guarantee is information-theoretically optimal up to a factor of $\log n$ and $\log(1/\eta)$.

In Appendix E.1, we set up polynomial systems that our algorithm uses and prove useful sos inequalities. In Appendix E.2, we show that we can easily obtain a robust algorithm via sos proofs in Appendix E.1. Then in Appendix E.3, we describe our algorithm and prove Theorem E.1.

## E.1 Sum-of-squares

For an adjacency matrix $A$ and nonnegative scalars $\gamma$, $\sigma$ and $\hat{d}$, consider the following polynomial systems with indeterminates $Y = (Y_{ij})_{i,j\in[n]}$, $z = (z_i)_{i\in[n]}$ and coefficients that depend on $A, \gamma, \sigma, \hat{d}$:

$$\mathcal{P}_1(Y, z; A, \gamma) := \begin{cases} z \odot z = z, \ \langle \mathbb{1}, z \rangle \geqslant (1-\gamma)n \\ 0 \leqslant Y \leqslant \mathbb{1}\mathbb{1}^\top, \ Y = Y^\top \\ Y \odot zz^\top = A \odot zz^\top \end{cases}, \tag{E.1}$$

$$\mathcal{P}_3(Y; \sigma, \hat{d}) := \left\{ (Y\mathbb{1})_i \leqslant \sigma\hat{d} \quad \forall i \in [n] \right\}. \tag{E.2}$$

For convenience of notation, we will consider the following combined polynomial system in the remaining of this section

$$\mathcal{D}(Y, z; A, \gamma, \sigma, \hat{d}) := \mathcal{P}_1(Y, z; A, \gamma) \cup \mathcal{P}_3(Y; \sigma, \hat{d}). \tag{E.3}$$

**Lemma E.2.** *If $(A^*, z^*)$ is a feasible solution to $\mathcal{D}(Y, z; A, \gamma^*, \sigma, \hat{d})$, then it follows that*

$$\mathcal{D}(Y, z; A, \gamma, \sigma, \hat{d}) \left|\frac{Y, z}{8} \ |d(Y) - d(A^*)| \leqslant 2(\gamma + \gamma^*)\sigma\hat{d}.\right.$$

*Proof.* Let $w = z \odot z^*$. Using similar analysis as in the proof of Lemma D.2, it follows that

$$\mathcal{D} \left|\frac{Y, z}{4} \ Y \odot ww^\top = A^* \odot ww^\top,\right.$$

and,

$$\begin{aligned} \mathcal{D} \left|\frac{Y, z}{8}\right. n \cdot d(Y) &= \langle Y, \mathbb{1}\mathbb{1}^\top \rangle \\ &= \langle A^*, ww^\top \rangle + \langle Y, 2(\mathbb{1}-w)\mathbb{1}^\top \rangle - \langle Y, (\mathbb{1}-w)(\mathbb{1}-w)^\top \rangle \\ &\leqslant \langle A^*, \mathbb{1}\mathbb{1}^\top \rangle + \langle Y\mathbb{1}, 2(\mathbb{1}-w) \rangle \\ &\leqslant n \cdot d(A^*) + 2(\gamma + \gamma^*)\sigma n\hat{d}. \end{aligned}$$

By rearranging the terms, we have

$$\mathcal{D} \left|\frac{Y, z}{8} \ d(Y) - d(A^*) \leqslant 2(\gamma + \gamma^*)\sigma\hat{d}.\right.$$

Swapping the roll of $Y$ and $A^*$, we can also get

$$\mathcal{D} \left|\frac{Y, z}{8} \ d(A^*) - d(Y) \leqslant 2(\gamma + \gamma^*)\sigma\hat{d}.\right.$$

This completes the proof. $\qquad\square$

**Lemma E.3.** *Let $Q^\circ$ be an n-by-n edge connection probability matrix and $d^\circ := d(Q^\circ)$. Suppose $\|Q^\circ\|_\infty \leqslant Rd^\circ/n$ for $R \in \mathbb{R}$. Let $A$ be an $\eta$-corrupted adjacency matrix of a random graph $G^\circ \sim \mathbb{G}(n, Q^\circ)$. Suppose $\eta\log(1/\eta)R \leqslant C_1$ for some constant $C_1$ that is small enough. With probability $1 - n^{-\Omega(1)}$, there exists $A^*$ and $z^*$ such that*

- $|d(A^*) - d^\circ| \leqslant 10\sqrt{\frac{d^\circ \log n}{n}} + 2\eta\log(1/\eta)Rd^\circ.$

- $(A^*, z^*)$ *is a feasible solution to* $\mathcal{D}(Y, z; A, \gamma, \sigma, \hat{d})$ *with $\eta$-corrupted $A$, $\gamma = 2\eta$, $\sigma = 10\log(1/\eta)R$ and $\hat{d} \geqslant \frac{1}{2}d^\circ$.*

*Proof.* Let $A^\circ$ be the adjacency matrix of $G^\circ$ and $z^\circ \in \{0, 1\}^n$ denote the set of uncorrupted nodes ($z_i^\circ = 1$ if and only if node $i$ is uncorrpted).

By Lemma B.2 and Lemma B.3, we know that, with probability $1 - n^{-\Omega(1)}$, there exists a degree-pruned adjacency matrix $\tilde{A}$ such that

1. $\left\|\tilde{A}\mathbb{1}\right\|_\infty \leqslant \log(1/\eta)Rd^\circ.$

2. At most $\eta n$ nodes are pruned.

3. At most $2\eta \log(1/\eta)nRd^\circ$ edges are pruned.

Let $\tilde{z} \in \{0,1\}^n$ denote the set of unpruned nodes ($\tilde{z}_i = 1$ if and only if node $i$ is not pruned). We will show that $A^* = \tilde{A}$ and $z^* = z^\circ \odot \tilde{z}$ satisfies the lemma.

**Guarantee 1.** By Lemma B.1, we know that, with probability $1 - n^{-\Omega(1)}$,

$$|d(A^\circ) - d^\circ| \leq 10\sqrt{\frac{d^\circ \log n}{n}}. \tag{E.4}$$

From degree pruning guarantee (3), we have that

$$|d(\tilde{A}) - d(A^\circ)| \leq 2\eta \log(1/\eta)Rd^\circ. \tag{E.5}$$

Combining Eq. (E.4) and Eq. (E.5), we have

$$|d(\tilde{A}) - d^\circ| \leq |d(\tilde{A}) - d(A^\circ)| + |d(A^\circ) - d^\circ|$$

$$\leq 10\sqrt{\frac{d^\circ \log n}{n}} + 2\eta \log(1/\eta)Rd^\circ.$$

**Guarantee 2.** It is easy to check that $z^* \odot z^* = z^*$, $0 \leq A^* \leq \mathbb{1}\mathbb{1}^\top$ and $A^* = (A^*)^\top$. Since $\langle \mathbb{1}, \tilde{z} \rangle \geq 1 - \eta n$ by degree pruning condition (2) and $\langle \mathbb{1}, z^\circ \rangle \geq 1 - \eta n$ by corruption rate, it is easy to verify that

$$\langle \mathbb{1}, z^* \rangle \geq 1 - 2\eta n.$$

Moreover, we have $A^* \odot z^*(z^*)^\top = A \odot z^*(z^*)^\top$ due to

$$\tilde{A} \odot \tilde{z}\tilde{z}^\top \odot z^\circ(z^\circ)^\top = A^\circ \odot \tilde{z}\tilde{z}^\top \odot z^\circ(z^\circ)^\top = A^\circ \odot z^\circ(z^\circ)^\top \odot \tilde{z}\tilde{z}^\top = A \odot z^\circ(z^\circ)^\top \odot \tilde{z}\tilde{z}^\top.$$

By degree pruning condition (1), we have

$$(A^*\mathbb{1})_i \leq \log(1/\eta)Rd^\circ \leq \sigma\hat{d}.$$

for all $i \in [n]$.

Thus, $(A^*, z^*)$ is a feasible solution to $\mathcal{D}(Y, z; A, \gamma, \sigma, \hat{d})$ with $\gamma = 2\eta$, $\sigma = 10\log(1/\eta)R$ and $\hat{d} \geq \frac{1}{2}d^\circ$. □

## E.2 Robust algorithm

In this section, we show that the following algorithm based on sum-of-squares proofs in Appendix E.1 obtains a robust approximation of $d^\circ$ that is optimal up to logarithmic factors.

---

**Algorithm E.4** (Robust fine estimation algorithm for inhomogeneous random graphs)**.**
**Input:** $\eta$-corrupted adjacency matrix $A$, corruption fraction $\eta$ and parameter $R$.

**Algorithm:**

1. Obtain coarse estimator $\hat{d}$ by applying Algorithm D.4 with $A, \eta, R$ as input.

2. Obtain level-8 pseudo-expectation $\tilde{\mathbb{E}}$ by solving sum-of-squares relaxation of program $\mathcal{D}(Y, z; A, \gamma, \sigma, \hat{d})$ (defined in Eq. (E.3)) with $A$, $\gamma = 2\eta$, $\sigma = 10\log(1/\eta)R$ and $\hat{d}$.

**Output:** $\tilde{\mathbb{E}}[d(Y)]$

---

**Theorem E.5** (Robust fine estimation for inhomogeneous random graphs)**.** *Let $Q^\circ$ be an n-by-n edge connection probability matrix and let $d^\circ := d(Q^\circ)$. Suppose $\|Q^\circ\|_\infty \leq Rd^\circ/n$ for some R. Let $A$ be an $\eta$-corrupted adjacency matrix of a random graph $G^\circ \sim \mathbb{G}(n, Q^\circ)$. Suppose*

$\eta \log(1/\eta)R \leqslant c$ *for some constant c that is small enough. With probability* $1 - n^{-\Omega(1)}$, *Algorithm E.4 outputs an estimate* $\tilde{d}$ *satisfying*

$$\left| \frac{\tilde{d}}{d^\circ} - 1 \right| \leqslant O\left( \sqrt{\frac{\log n}{d^\circ n}} + \eta \log(1/\eta)R \right).$$

*Proof.* By Theorem D.5, we have $\frac{1}{2}d^\circ \leqslant \hat{d} \leqslant \frac{3}{2}d^\circ$. Let $\gamma^* = 2\eta$, by Lemma E.2 and Lemma E.3, it follows that

$$\mathcal{D}(Y, z; A, \gamma, \sigma, \hat{d}) \Big|_{8}^{Y,z} |d(Y) - d(A^*)| \leqslant 2(\gamma + \gamma^*)\sigma\hat{d}$$

$$\leqslant 2 \cdot 4\eta \cdot 10 \log(1/\eta)R \cdot \frac{3}{2}d^\circ$$

$$= 120\eta \log(1/\eta)Rd^\circ .$$

and,

$$|d(A^*) - d^\circ| \leqslant 10\sqrt{\frac{d^\circ \log n}{n}} + 2\eta \log(1/\eta)Rd^\circ .$$

Therefore, we have

$$\mathcal{D}(Y, z; A, \gamma, \sigma, \hat{d}) \Big|_{O(1)}^{Y,z} |d(Y) - d^\circ| \leqslant 200\eta \log(1/\eta)Rd^\circ + 10\sqrt{\frac{d^\circ \log n}{n}} .$$

Thus, the level-8 pseudo-expectation $\tilde{\mathbb{E}}$ satisfies

$$\left| \tilde{\mathbb{E}}[d(Y)] - d^\circ \right| \leqslant 200\eta \log(1/\eta)Rd^\circ + 10\sqrt{\frac{d^\circ \log n}{n}} ,$$

which implies that

$$\left| \frac{\tilde{\mathbb{E}}[d(Y)]}{d^\circ} - 1 \right| \leqslant O\left( \sqrt{\frac{\log n}{d^\circ n}} + \eta \log(1/\eta)R \right).$$

$\square$

### E.3 Private algorithm

In this section, we present our algorithm and prove Theorem E.1. Our algorithm instantiates the sum-of-squares exponential mechanism in Appendix C.

**Score function.** For an $n$-by-$n$ symmetric matrix $A$ and a scalar $d$, we define the score of $d$ with regard to $A$ to be

$$s(d; A) := \min_{0 \leqslant \gamma \leqslant 1} \gamma n \text{ s.t.} \begin{cases} \exists \text{ level-8 pseudo-expectation } \tilde{\mathbb{E}} \text{ satisfying} \\ \mathcal{D}(Y, z; A, \gamma, \sigma, \hat{d}) \cup \{|d(Y) - d| \leqslant \alpha d\} , \end{cases} \tag{E.6}$$

where $\mathcal{D}(Y, z; A, \gamma, \sigma, \hat{d})$ is the polynomial system defined in Eq. (E.3), $\hat{d}$ is a coarse estimate, and $\sigma$, $\alpha$ are fixed parameters whose values will be decided later. Note that $(Y = \frac{d}{n}\mathbb{1}\mathbb{1}^\top, z = \mathbb{0})$ is a solution to the polynomial system $\mathcal{D}(Y, z; A, 1, \sigma, \hat{d}) \cup \{|d(Y) - d| \leqslant \alpha d\}$ for any $A \in \mathbb{R}^{n \times n}$ and any $d$ such that $0 \leqslant d \leqslant \min\{\sigma\hat{d}, n\}$.

To efficiently compute $s(d; A)$, we can use the scheme as described in Remark C.1.

**Exponential mechanism.** Given a privacy parameter $\varepsilon$ and an $n$-by-$n$ symmetric matrix $A$, our private algorithm in Theorem E.1 is the exponential mechanism with score function Eq. (E.6) and range $[0, \min\{\sigma\hat{d}, n\}]$.

> **Algorithm E.6** (Fine estimation for inhomogeneous random graphs).
> **Input:** Graph $A$, coarse estimate $\hat{d}$.
>
> **Parameters:** $\varepsilon, \sigma, \alpha$.
>
> **Output:** A sample from the distribution $\mu_{A,\varepsilon}$ with support $[0, \min\{\sigma\hat{d}, n\}]$ and density
> $$\mathrm{d}\mu_{A,\varepsilon}(d) \propto \exp(-\varepsilon \cdot s(d; A)), \tag{E.7}$$
> where $s(d; A)$ is defined in Eq. (E.6).

To efficiently sample from $\mu_{A,\varepsilon}$, we can use the scheme as described in Remark C.2.

**Privacy.** The following privacy guarantee of our algorithm is a direct corollary of Lemma C.4.

**Lemma E.7** (Privacy). *Algorithm E.6 is $2\varepsilon$-differentially node private.*

**Utility.** The utility guarantee of our algorithm is stated in the following lemma.

**Lemma E.8** (Utility). *Let $Q^\circ$ be an $n$-by-$n$ edge connection probability matrix and let $d^\circ := d(Q^\circ)$. Suppose $\|Q^\circ\|_\infty \leqslant Rd^\circ/n$ for some $R$. There is a sufficiently small constant $c$ such that the following holds. For any $\eta$ such that $\eta \log(1/\eta)R \leqslant c$, given an $\eta$-corrupted inhomogeneous random graph $\mathbb{G}(n, Q^\circ)$ and a coarse estimate $\hat{d}$ such that $0.5d^\circ \leqslant \hat{d} \leqslant 2d^\circ$, Algorithm E.6 outputs an estimate $\tilde{d}$ satisfying*
$$\left| \frac{\tilde{d}}{d^\circ} - 1 \right| \leqslant O\left( \sqrt{\frac{\log n}{d^\circ n}} + \frac{R \log^2 n}{\varepsilon n} + \eta \log(1/\eta)R \right),$$
*with probability $1 - n^{-\Omega(1)}$.*

Before proving Lemma E.8, we need the following two lemmas.

**Lemma E.9** (Volume of low-score points). *Let $A \in \mathbb{R}^{n \times n}$ and $\varepsilon > 0$. Consider the distribution $\mu_{A,\varepsilon}$ defined by Eq. (E.7). Suppose $(Y = A^*, z = z^*)$ is a solution to $\mathcal{D}(Y, z; A, \gamma^*, \sigma, \hat{d})$ and $2 \leqslant d(A^*) \leqslant \sigma\hat{d}$. Then for any $t \geqslant 0$,*
$$\mathbb{P}_{d \sim \mu_{A,\varepsilon}}\left( s(d; A) \geqslant \gamma^* n + \frac{t \log n}{\varepsilon} \right) \leqslant \frac{n^{-t+1}}{\alpha}.$$

*Proof.* Apply Lemma C.5 with $\mathcal{D} = [0, \min\{\sigma\hat{d}, n\}]$ and
$$\mathcal{G}(A^*) = \left\{ d \in \mathcal{D} \; : \; \frac{d(A^*)}{1+\alpha} \leqslant d \leqslant \frac{d(A^*)}{1-\alpha} \right\}.$$
As $[d(A^*)/(1+\alpha), d(A^*)] \subseteq \mathcal{G}(A^*)$ and $d(A^*) \geqslant 2 \geqslant 1 + \alpha$, we have $\mathrm{vol}(\mathcal{G}(A^*)) \geqslant \alpha$. $\qquad\square$

**Lemma E.10** (Low score implies utility). *Let $A \in \mathbb{R}^{n \times n}$ and consider the score function $s(\cdot; A)$ defined in Eq. (E.6). Suppose $(Y = A^*, z = z^*)$ is a solution to $\mathcal{D}(Y, z; A, \gamma^*, \sigma, \hat{d})$. For a scalar $d$ such that $s(d; A) \leqslant \tau n$,*
$$\frac{d(A^*) - 2(\gamma^* + \tau)\sigma\hat{d}}{1+\alpha} \leqslant d \leqslant \frac{d(A^*) + 2(\gamma^* + \tau)\sigma\hat{d}}{1-\alpha}.$$

*Proof.* By Lemma E.2,
$$\mathcal{D}(Y, z; A, \tau, \sigma, \hat{d}) \left|\begin{smallmatrix} Y,z \\ 8 \end{smallmatrix}\right. |d(Y) - d(A^*)| \leqslant 2(\gamma^* + \tau)\sigma\hat{d}.$$
Thus,
$$\mathcal{D}(Y, z; A, \tau, \sigma, \hat{d}) \cup \{|d(Y) - d| \leqslant \alpha d\}$$
$$\left|\begin{smallmatrix} Y,z \\ 8 \end{smallmatrix}\right. \frac{d(A^*) - 2(\gamma^* + \tau)\sigma\hat{d}}{1+\alpha} \leqslant d \leqslant \frac{d(A^*) + 2(\gamma^* + \tau)\sigma\hat{d}}{1-\alpha}.$$
$\qquad\square$

Now we are ready to prove Lemma E.8.

*Proof of Lemma E.8.* Let $A$ be a realization of $\eta$-corrupted $\mathbb{G}(n, Q^\circ)$. By Lemma E.3, the following event happens with probability at least $1 - n^{-\Omega(1)}$. There exists a solution $(Y = A^*, z = z^*)$ to $\mathcal{D}(Y, z; A, \gamma^*, \sigma, \hat{d})$ with $\gamma^* = 2\eta$, $\sigma = 10 \log(1/\eta)R$, and

$$|d(A^*) - d^\circ| \leq 10\sqrt{d^\circ \log(n)/n} + 2\eta \log(1/\eta)Rd^\circ.$$

For $\eta \log(1/\eta)R$ smaller than some constant, we have $0.9d^\circ \leq d(A^*) \leq 1.1d^\circ$. Note that $d(A^*) \geq 0.9d^\circ \geq 2$ and $d(A^*) \leq 1.1d^\circ \leq \sigma\hat{d}$. Then it follows by setting $t = 10$ and $\alpha = n^{-2}$ in Lemma E.9 that,

$$\mathbb{P}_{d \sim \mu_{A,\varepsilon}}(s(d; A) \leq \tau n) \geq 1 - n^{-7} \text{ where } \tau := 2\eta + 10 \log(n)/(\varepsilon n).$$

Let $\tilde{d}$ be a scalar such that $s(\tilde{d}; A) \leq \tau n$. Then by Lemma E.10,

$$\frac{d(A^*) - 2(2\eta + \tau)\sigma\hat{d}}{1 + \alpha} \leq \tilde{d} \leq \frac{d(A^*) + 2(2\eta + \tau)\sigma\hat{d}}{1 - \alpha}.$$

Plugging in everything, we have

$$\left|\frac{\tilde{d}}{d^\circ} - 1\right| \leq O\left(\sqrt{\frac{\log n}{d^\circ n}} + \frac{R \log(1/\eta) \log n}{\varepsilon n} + R\eta \log(1/\eta)\right).$$

As an $\eta$-corrupted graph is actually uncorrupted when $\eta < 1/n$, we can assume $\eta \geq 1/(2n)$ without loss of generality. Therefore,

$$\left|\frac{\tilde{d}}{d^\circ} - 1\right| \leq O\left(\sqrt{\frac{\log n}{d^\circ n}} + \frac{R \log^2 n}{\varepsilon n} + R\eta \log(1/\eta)\right).$$

$\square$

**Proof of Theorem E.1.** By Lemma E.7 and Lemma E.8.

# F   Fine estimation for Erdős-Rényi random graphs

From Appendix D, we have a a constant multiplicative approximation of the expected average degree $d^\circ$. In this section, we show how to use this coarse estimate to obtain our fine estimate for Erdős-Rényi random graphs.

**Theorem F.1** (Fine estimation for Erdős-Rényi random graphs)**.** *There are constants $C_1, C_2, C_3$ such that the following holds. For any $\eta \leq C_1$, $\varepsilon \geq C_2 \log(n)/n$, and $d^\circ \geq C_3$, there exists a polynomial-time $\varepsilon$-differentially node private algorithm which, given an $\eta$-corrupted Erdős-Rényi random graph $\mathbb{G}(n, d^\circ/n)$ and a constant-factor approximation of $d^\circ$, outputs an estimate $\tilde{d}$ satisfying*

$$\left|\frac{\tilde{d}}{d^\circ} - 1\right| \leq O\left(\sqrt{\frac{\log n}{d^\circ n}} + \frac{\log^2 n}{\sqrt{d^\circ}\varepsilon n} + \frac{\eta \log n}{\sqrt{d^\circ}}\right),$$

*with probability $1 - n^{-\Omega(1)}$.*

We make a few remarks on Theorem F.1.

- Our algorithm in Theorem F.1 is an sum-of-squares exponential mechanism. $R, \eta, \varepsilon$ are parameters given as input to our algorithm.

- We can get an estimate of $p^\circ$ by taking $\hat{p} = \frac{\hat{d}}{n-1}$. Since $\frac{\hat{p}}{p^\circ} = \frac{\hat{d}}{d^\circ}$, it follows that

$$\left|\frac{\hat{p}}{p^\circ} - 1\right| \leq O\left(\sqrt{\frac{\log n}{d^\circ n}} + \frac{\log^2 n}{\sqrt{d^\circ}\varepsilon n} + \frac{\eta \log n}{\sqrt{d^\circ}}\right).$$

- Combining [Theorem D.1](#) and [Theorem F.1](#) gives us an efficient, private, and robust edge density estimation algorithm for Erdős-Rényi random graphs whose utility guarantee is information-theoretically optimal up to a factor of $\log n$.

In [Appendix F.1](#), we set up polynomial systems that our algorithm uses and prove useful sos inequalities. In [Appendix F.2](#), we show that we can easily obtain a robust algorithm via sos proofs in [Appendix F.1](#). Then in [Appendix F.3](#), we describe our algorithm and prove [Theorem F.1](#).

### F.1 Sum-of-squares

For an adjacency matrix $A$ and nonnegative scalars $\gamma$, $\sigma$ and $\hat{d}$, consider the following polynomial systems with indeterminates $Y = (Y_{ij})_{i,j \in [n]}$, $z = (z_i)_{i \in [n]}$ and coefficients that depend on $A, \gamma, \sigma, \delta, \hat{d}$:

$$\mathcal{P}_1(Y, z; A, \gamma) := \begin{cases} z \odot z = z, \ \langle \mathbb{1}, z \rangle \geq (1 - \gamma)n \\ 0 \leq Y \leq \mathbb{1}\mathbb{1}^\top, \ Y = Y^\top \\ Y \odot zz^\top = A \odot zz^\top \end{cases}, \tag{F.1}$$

$$\mathcal{P}_4(Y; \sigma, \delta, \hat{d}) := \begin{cases} |(Y\mathbb{1})_i - d(Y)| \leq \sigma\sqrt{\hat{d}} & \forall i \in [n] \\ \left\| Y - \frac{d(Y)}{n}\mathbb{1}\mathbb{1}^\top \right\|_{\mathrm{op}} \leq \delta\sqrt{\hat{d}} \end{cases}. \tag{F.2}$$

For convenience of notation, we will consider the following combined polynomial system in remaining of the section

$$\mathcal{E}(Y, z; A, \gamma, \sigma, \delta, \hat{d}) := \mathcal{P}_1(Y, z; A, \gamma) \cup \mathcal{P}_4(Y; \sigma, \delta, \hat{d}). \tag{F.3}$$

**Lemma F.2.** *If $(A^*, z^*)$ is a feasible solution to $\mathcal{E}(Y, z; A, \gamma^*, \sigma, \delta, \hat{d})$ and $\gamma + \gamma^* < 1$, then it follows that*

$$\mathcal{E}(Y, z; A, \gamma, \sigma, \delta, \hat{d}) \left|\frac{Y, z}{8}\right. |d(Y) - d(A^*)| \leq \frac{4(\gamma + \gamma^*)\sigma\sqrt{\hat{d}} + 2(\gamma + \gamma^*)\delta\sqrt{\hat{d}}}{(1 - \gamma - \gamma^*)^2}.$$

*Proof.* Let $w = z \odot z^*$. Notice that, by [Lemma A.9](#), we have $\mathcal{E} \left|\frac{z}{4}\right. 1 - w_i \leq 2 - z_i - z_i^*$ for all $i \in [n]$. Moreover, using similar analysis as in the proof of [Lemma D.2](#), it follows that

$$\mathcal{E} \left|\frac{Y, z}{4}\right. Y \odot ww^\top = A^* \odot ww^\top.$$

Therefore, we can get

$$
\begin{aligned}
\mathcal{E} \left|\frac{Y, z}{4}\right. n\big(d(Y) - d(A^*)\big) &= \langle Y - A^*, \mathbb{1}\mathbb{1}^\top \rangle \\
&= \langle Y - A^*, \mathbb{1}\mathbb{1}^\top - ww^\top \rangle \\
&= \langle Y - \frac{d(Y)}{n}\mathbb{1}\mathbb{1}^\top + \frac{d(Y)}{n}\mathbb{1}\mathbb{1}^\top - \frac{d(A^*)}{n}\mathbb{1}\mathbb{1}^\top + \frac{d(A^*)}{n}\mathbb{1}\mathbb{1}^\top - A^*, \mathbb{1}\mathbb{1}^\top - ww^\top \rangle \\
&= \langle Y - \frac{d(Y)}{n}\mathbb{1}\mathbb{1}^\top, \mathbb{1}\mathbb{1}^\top - ww^\top \rangle + \langle \frac{d(A^*)}{n}\mathbb{1}\mathbb{1}^\top - A^*, \mathbb{1}\mathbb{1}^\top - ww^\top \rangle \\
&\quad + \langle \frac{d(Y)}{n}\mathbb{1}\mathbb{1}^\top - \frac{d(A^*)}{n}\mathbb{1}\mathbb{1}^\top, \mathbb{1}\mathbb{1}^\top - ww^\top \rangle \\
&= \langle Y - \frac{d(Y)}{n}\mathbb{1}\mathbb{1}^\top, \mathbb{1}\mathbb{1}^\top - ww^\top \rangle + \langle \frac{d(A^*)}{n}\mathbb{1}\mathbb{1}^\top - A^*, \mathbb{1}\mathbb{1}^\top - ww^\top \rangle \\
&\quad + \big(d(Y) - d(A^*)\big)\Big(n - \frac{1}{n}\langle \mathbb{1}, w \rangle^2\Big).
\end{aligned}
\tag{F.4}
$$

By rearranging terms, we can get

$$\mathcal{E} \left|\frac{Y, z}{8}\right. \frac{\langle \mathbb{1}, w \rangle^2}{n}\big(d(Y) - d(A^*)\big) = \langle Y - \frac{d(Y)}{n}\mathbb{1}\mathbb{1}^\top, \mathbb{1}\mathbb{1}^\top - ww^\top \rangle + \langle \frac{d(A^*)}{n}\mathbb{1}\mathbb{1}^\top - A^*, \mathbb{1}\mathbb{1}^\top - ww^\top \rangle. \tag{F.5}$$

We bound the two terms on the right-hand side separately. For the first term $\langle Y - \frac{d(Y)}{n} \mathbb{1}\mathbb{1}^\top, \mathbb{1}\mathbb{1}^\top - ww^\top \rangle$, we have

$$\mathcal{E}\left|\frac{Y,z}{8}\right. \langle Y - \frac{d(Y)}{n}\mathbb{1}\mathbb{1}^\top, \mathbb{1}\mathbb{1}^\top - ww^\top \rangle = 2\langle Y - \frac{d(Y)}{n}\mathbb{1}\mathbb{1}^\top, \mathbb{1}(\mathbb{1}-w)^\top \rangle + \langle \frac{d(Y)}{n}\mathbb{1}\mathbb{1}^\top - Y, (\mathbb{1}-w)(\mathbb{1}-w)^\top \rangle. \tag{F.6}$$

From constraints $|(Y\mathbb{1})_i - d(Y)| \leqslant \sigma\sqrt{\hat{d}}$ for all $i \in [n]$, $\langle \mathbb{1}, z \rangle \geqslant (1-\gamma)n$ and $\langle \mathbb{1}, z^* \rangle \geqslant (1-\gamma^*)n$, we have

$$\begin{aligned}
\mathcal{E}\left|\frac{Y,z}{8}\right. \langle Y - \frac{d(Y)}{n}\mathbb{1}\mathbb{1}^\top, \mathbb{1}(\mathbb{1}-w)^\top \rangle &= \langle Y\mathbb{1} - d(Y)\mathbb{1}, \mathbb{1} - w \rangle \\
&\leqslant \sum_{i \in [n]} (1 - w_i)\sigma\sqrt{\hat{d}} \\
&\leqslant \sigma\sqrt{\hat{d}} \cdot \left( \sum_{i \in [n]} 2 - z_i - z_i^* \right) \\
&\leqslant (\gamma + \gamma^*)n\sigma\sqrt{\hat{d}}.
\end{aligned} \tag{F.7}$$

From constraints $\left\| Y - \frac{d(Y)}{n}\mathbb{1}\mathbb{1}^\top \right\|_{\text{op}} \leqslant \delta\sqrt{\hat{d}}$, $\langle \mathbb{1}, z \rangle \geqslant (1-\gamma)n$ and $\langle \mathbb{1}, z^* \rangle \geqslant (1-\gamma^*)n$, we have

$$\begin{aligned}
\mathcal{E}\left|\frac{Y,z}{8}\right. \langle \frac{d(Y)}{n}\mathbb{1}\mathbb{1}^\top - Y, (\mathbb{1}-w)(\mathbb{1}-w)^\top \rangle &\leqslant \left\| Y - \frac{d(Y)}{n}\mathbb{1}\mathbb{1}^\top \right\|_{\text{op}} \|\mathbb{1} - w\|_2^2 \\
&\leqslant \delta\sqrt{\hat{d}} \cdot \left( \sum_{i \in [n]} (1 - w_i)^2 \right) \\
&= \delta\sqrt{\hat{d}} \cdot \left( \sum_{i \in [n]} 1 - w_i \right) \\
&\leqslant \delta\sqrt{\hat{d}} \cdot \left( \sum_{i \in [n]} 2 - z_i - z_i^* \right) \\
&\leqslant (\gamma + \gamma^*)n\delta\sqrt{\hat{d}},
\end{aligned} \tag{F.8}$$

where the equality is because $\mathcal{E}\left|\frac{z}{2}\right. (1 - w_i)^2 = (1 - z_i z_i^*)^2 = 1 - z_i z_i^* = 1 - w_i$.

Plugging Eq. (F.7) and Eq. (F.8) into Eq. (F.6), it follows that

$$\mathcal{E}\left|\frac{Y,z}{8}\right. \langle Y - \frac{d(Y)}{n}\mathbb{1}\mathbb{1}^\top, \mathbb{1}\mathbb{1}^\top - ww^\top \rangle \leqslant 2(\gamma + \gamma^*)n\sigma\sqrt{\hat{d}} + (\gamma + \gamma^*)n\delta\sqrt{\hat{d}}. \tag{F.9}$$

For the second term $\langle \frac{d(A^*)}{n}\mathbb{1}\mathbb{1}^\top - A^*, \mathbb{1}\mathbb{1}^\top - ww^\top \rangle$, we can apply the same proof as above to get

$$\mathcal{E}\left|\frac{Y,z}{8}\right. \langle \frac{d(A^*)}{n}\mathbb{1}\mathbb{1}^\top - A^*, \mathbb{1}\mathbb{1}^\top - ww^\top \rangle \leqslant 2(\gamma + \gamma^*)n\sigma\sqrt{\hat{d}} + (\gamma + \gamma^*)n\delta\sqrt{\hat{d}}. \tag{F.10}$$

Plugging Eq. (F.9) and Eq. (F.10) into Eq. (F.5), it follows that

$$\mathcal{E}\left|\frac{Y,z}{8}\right. \frac{\langle \mathbb{1}, w \rangle^2}{n}\left( d(Y) - d(A^*) \right) \leqslant 4(\gamma + \gamma^*)n\sigma\sqrt{\hat{d}} + 2(\gamma + \gamma^*)n\delta\sqrt{\hat{d}}.$$

Using the same proof strategy, we can also get

$$\mathcal{E}\left|\frac{Y,z}{8}\right. \frac{\langle \mathbb{1}, w \rangle^2}{n}\left( d(A^*) - d(Y) \right) \leqslant 4(\gamma + \gamma^*)n\sigma\sqrt{\hat{d}} + 2(\gamma + \gamma^*)n\delta\sqrt{\hat{d}}.$$

Applying Lemma A.10, it follows that

$$\mathcal{E}\left|\frac{Y,z}{8}\right. \frac{\langle \mathbb{1}, w\rangle^4}{n^2}\left(d(Y) - d(A^*)\right)^2 \leqslant \left(4(\gamma + \gamma^*)n\sigma\sqrt{\hat{d}} + 2(\gamma + \gamma^*)n\delta\sqrt{\hat{d}}\right)^2 . \tag{F.11}$$

Now, we would like to lower bound $\langle \mathbb{1}, w\rangle^4$. By Lemma A.9, we have $\mathcal{E}\left|\frac{z}{4}\right. w_i \geqslant z_i + z_i^* - 1$ for all $i \in [n]$. Therefore,

$$\mathcal{E}\left|\frac{Y,z}{8}\right. \langle \mathbb{1}, w\rangle = \sum_{i\in[n]} w_i \geqslant \sum_{i\in[n]}(z_i + z_i^* - 1) \geqslant (1 - \gamma - \gamma^*)n .$$

Since $\gamma + \gamma^* < 1$, we have $1 - \gamma - \gamma^* > 0$, and, therefore,

$$\mathcal{E}\left|\frac{Y,z}{8}\right. \langle \mathbb{1}, w\rangle^4 \geqslant (1 - \gamma - \gamma^*)^4 n^4 .$$

Plugging this into Eq. (F.11), we have

$$\mathcal{E}\left|\frac{Y,z}{8}\right. (1 - \gamma - \gamma^*)^4 n^2\left(d(Y) - d(A^*)\right)^2 \leqslant \left(4(\gamma + \gamma^*)n\sigma\sqrt{\hat{d}} + 2(\gamma + \gamma^*)n\delta\sqrt{\hat{d}}\right)^2$$

$$\left|\frac{Y,z}{8}\right. \left(d(Y) - d(A^*)\right)^2 \leqslant \frac{\left(4(\gamma + \gamma^*)\sigma\sqrt{\hat{d}} + 2(\gamma + \gamma^*)\delta\sqrt{\hat{d}}\right)^2}{(1 - \gamma - \gamma^*)^4} .$$

Applying Lemma A.11, it follows that

$$\mathcal{E}\left|\frac{Y,z}{8}\right. |d(Y) - d(A^*)| \leqslant \frac{4(\gamma + \gamma^*)\sigma\sqrt{\hat{d}} + 2(\gamma + \gamma^*)\delta\sqrt{\hat{d}}}{(1 - \gamma - \gamma^*)^2} .$$

$\square$

**Lemma F.3.** *Let $A$ be an $\eta$-corrupted adjacency matrix of a random graph $G^\circ \sim \mathbb{G}(n, \frac{d^\circ}{n})$. With probability $1 - n^{-\Omega(1)}$, there exists $A^*$ and $z^*$ such that*

- $|d(A^*) - d^\circ| \leqslant 10\sqrt{\frac{d^\circ \log n}{n}}$.

- $(A^*, z^*)$ *is a feasible solution to $\mathcal{E}(Y, z; A, \gamma, \sigma, \delta, \hat{d})$ with $\gamma = \eta$, $\sigma = 4\log n$, $\delta = 4C\sqrt{\log n}$ for some constant $C$ and $\hat{d} \geqslant \frac{1}{2}d^\circ$.*

*Proof.* Let $A^\circ$ be the adjacency matrix of $G^\circ$ and $z^\circ \in \{0, 1\}^n$ denote the set of uncorrupted nodes ($z_i^\circ = 1$ if and only if node $i$ is uncorrupted). We will show that $A^* = A^\circ$ and $z^* = z^\circ$ satisfies the lemma.

**Guarantee 1.** By Lemma B.1, we know that, with probability $1 - n^{-\Omega(1)}$,

$$|d(A^\circ) - d^\circ| \leqslant 10\sqrt{\frac{d^\circ \log n}{n}} . \tag{F.12}$$

**Guarantee 2.** It is easy to check that $z^* \odot z^* = z^*$, $0 \leqslant A^* \leqslant \mathbb{1}\mathbb{1}^\top$, $A^* = (A^*)^\top$ and $\langle \mathbb{1}, z^*\rangle \geqslant 1 - \eta n$. By Lemma B.2, we know that, with probability $1 - n^{-\Omega(1)}$,

$$\|A^\circ\mathbb{1} - d^\circ\mathbb{1}\|_\infty \leqslant \sqrt{d^\circ}\log n . \tag{F.13}$$

Combining Eq. (F.12) and Eq. (F.13), we have

$$\|A^\circ\mathbb{1} - d(A^\circ)\mathbb{1}\|_\infty \leqslant \|A^\circ\mathbb{1} - d^\circ\mathbb{1}\|_\infty + \|d^\circ\mathbb{1} - d(A^\circ)\mathbb{1}\|_\infty$$

$$\leqslant \sqrt{d^\circ}\log n + 10\sqrt{\frac{d^\circ \log n}{n}} \tag{F.14}$$

$$\leqslant 2\log n\sqrt{d^\circ} .$$

Therefore, for $\sigma = 4 \log n$ and $\hat{d} \geqslant \frac{1}{2} d^\circ$, it follows that

$$|(A^* \mathbb{1})_i - d(A^*)| \leqslant 2 \log n \sqrt{d^\circ} \leqslant \sigma \sqrt{\hat{d}},$$

for all $i \in [n]$.

By Lemma B.5, we know that, with probability $1 - n^{-\Omega(1)}$, for some universal constant $C$,

$$\left\| A^\circ - \frac{d^\circ}{n} \mathbb{1} \mathbb{1}^\top \right\|_{\mathrm{op}} \leqslant C \sqrt{d^\circ \log n}. \tag{F.15}$$

Combining Eq. (F.12) and Eq. (F.15), we have

$$
\begin{aligned}
\left\| A^\circ - \frac{d(A^\circ)}{n} \mathbb{1} \mathbb{1}^\top \right\|_{\mathrm{op}} &\leqslant \left\| A^\circ - \frac{d(A^\circ)}{n} \mathbb{1} \mathbb{1}^\top \right\|_{\mathrm{op}} + \left\| \frac{d(A^\circ)}{n} \mathbb{1} \mathbb{1}^\top - \frac{d^\circ}{n} \mathbb{1} \mathbb{1}^\top \right\|_{\mathrm{op}} \\
&\leqslant C \sqrt{d^\circ \log n} + 10 \sqrt{\frac{d^\circ \log n}{n}} \\
&\leqslant 2C \sqrt{d^\circ \log n}.
\end{aligned}
\tag{F.16}
$$

Therefore, for $\delta = 4C \sqrt{\log n}$ and $\hat{d} \geqslant \frac{1}{2} d^\circ$, it follows that

$$\left\| A^* - \frac{d(A^*)}{n} \mathbb{1} \mathbb{1}^\top \right\|_{\mathrm{op}} \leqslant 2C \sqrt{d^\circ \log n} \leqslant \delta \sqrt{\hat{d}}.$$

Thus, $(A^*, z^*)$ is a feasible solution to $\mathcal{E}(Y, z; A, \gamma, \sigma, \delta, \hat{d})$ with $\gamma = \eta$, $\sigma = 4 \log n$, $\delta = 4C \sqrt{\log n}$ and $\hat{d} \geqslant \frac{1}{2} d^\circ$. $\qquad\square$

## F.2 Robust algorithm

In this section, we show that the following algorithm based on sum-of-squares proofs in Appendix F.1 obtains a robust approximation of $d^\circ$ that is optimal up to logarithmic factors.

---

**Algorithm F.4** (Robust fine estimation algorithm for Erdős-Rényi random graphs).
**Input:** $\eta$-corrupted adjacency matrix $A$ and corruption fraction $\eta$.

**Algorithm:**

1. Obtain coarse estimator $\hat{d}$ by applying Algorithm D.4 with $A, \eta, R = 1$ as input.

2. Obtain level-8 pseudo-expectation $\tilde{\mathbb{E}}$ by solving sum-of-squares relaxation of program $\mathcal{E}(Y, z; A, \gamma, \sigma, \delta, \hat{d})$ (defined in Eq. (F.3)) with $A$, $\gamma = \eta$, $\sigma = 4 \log n$, $\delta = 4C \sqrt{\log n}$ and $\hat{d}$.

**Output:** $\tilde{\mathbb{E}}[d(Y)]$

---

**Theorem F.5** (Robust fine estimation for Erdős-Rényi random graphs). *Let $A$ be an $\eta$-corrupted adjacency matrix of a random graph $G^\circ \sim \mathbb{G}(n, \frac{d^\circ}{n})$. With probability $1 - n^{-\Omega(1)}$, Algorithm F.4 outputs an estimate $\tilde{d}$ satisfying*

$$\left| \frac{\tilde{d}}{d^\circ} - 1 \right| \leqslant O\left( \sqrt{\frac{\log n}{d^\circ n}} + \frac{\eta \log n}{\sqrt{d^\circ}} \right).$$

*Proof.* By Theorem D.5, we have $\frac{1}{2} d^\circ \leqslant \hat{d} \leqslant \frac{3}{2} d^\circ$. Let $\gamma^* = \eta$, by Lemma F.2 and Lemma F.3, it follows that

$$\mathcal{E}(Y, z; A, \gamma, \sigma, \delta, \hat{d}) \Big|_{O(1)}^{Y, z} |d(Y) - d(A^*)| \leqslant \frac{4(\gamma + \gamma^*) \sigma \sqrt{\hat{d}} + 2(\gamma + \gamma^*) \delta \sqrt{\hat{d}}}{(1 - \gamma - \gamma^*)^2}$$

$$\leqslant \frac{40\eta \log n \sqrt{d^\circ} + 40C\eta \sqrt{d^\circ \log n}}{(1 - 2\eta)^2}$$

$$\leqslant C'\eta \log n \sqrt{d^\circ} \,,$$

for some constant $C'$, and,

$$|d(A^*) - d^\circ| \leqslant 10\sqrt{\frac{d^\circ \log n}{n}} \,.$$

Therefore, we have

$$\mathcal{E}(Y, z; A, \gamma, \sigma, \delta, \hat{d})\Big|_{O(1)}^{Y,z} |d(Y) - d^\circ| \leqslant C'\eta \log n \sqrt{d^\circ} + 10\sqrt{\frac{d^\circ \log n}{n}} \,.$$

Thus, the level-8 pseudo-expectation $\tilde{\mathbb{E}}$ satisfies

$$\left|\tilde{\mathbb{E}}[d(Y)] - d^\circ\right| \leqslant C'\eta \log n \sqrt{d^\circ} + 10\sqrt{\frac{d^\circ \log n}{n}} \,,$$

which implies that

$$\left|\frac{\tilde{\mathbb{E}}[d(Y)]}{d^\circ} - 1\right| O\left(\sqrt{\frac{\log n}{d^\circ n}} + \frac{\eta \log n}{\sqrt{d^\circ}}\right).$$

$\square$

### F.3 Private algorithm

In this section, we present our algorithm and prove Theorem F.1. Our algorithm instantiates the sum-of-squares exponential mechanism in Appendix C.

**Score function.** For an $n$-by-$n$ symmetric matrix $A$ and a scalar $d$, we define the score of $d$ with regard to $A$ to be

$$s(d; A) := \min_{0 \leqslant \gamma \leqslant 1} \gamma n \text{ s.t. } \begin{cases} \exists \text{ level-8 pseudo-expectation } \tilde{\mathbb{E}} \text{ satisfying} \\ \mathcal{E}(Y, z; A, \gamma, \sigma, \delta, \hat{d}) \cup \{|d(Y) - d| \leqslant \alpha d\} \,, \end{cases} \tag{F.17}$$

where $\mathcal{E}(Y, z; A, \gamma, \sigma, \delta, \hat{d})$ is the polynomial system defined in Eq. (F.3), $\hat{d}$ is a coarse estimate, and $\sigma, \delta, \alpha$ are fixed parameters whose values will be decided later. Note that $(Y = \frac{d}{n}\mathbb{1}\mathbb{1}^\top, z = \mathbb{0})$ is a solution to the polynomial system $\mathcal{E}(Y, z; A, 1, \sigma, \delta, \hat{d}) \cup \{|d(Y)/d - 1| \leqslant \alpha\}$ for any $A \in \mathbb{R}^{n \times n}$ and any $d \in [0, n]$.

To efficiently compute $s(d; A)$, we can use the scheme as described in Remark C.1.

**Exponential mechanism.** Given a privacy parameter $\varepsilon$ and an $n$-by-$n$ symmetric matrix $A$, our private algorithm in Theorem F.1 is the exponential mechanism with score function Eq. (F.17) and range $[0, n]$.

---

**Algorithm F.6** (Fine estimation for Erdős-Rényi random graphs)**.**
**Input:** Graph $A$, coarse estimate $\hat{d}$.

**Parameters:** $\varepsilon, \sigma, \delta, \alpha$.

**Output:** A sample from the distribution $\mu_{A,\varepsilon}$ with support $[0, n]$ and density

$$d\mu_{A,\varepsilon}(d) \propto \exp(-\varepsilon \cdot s(d; A)) \,, \tag{F.18}$$

where $s(d; A)$ is defined in Eq. (F.17).

---

To efficiently sample from $\mu_{A,\varepsilon}$, we can use the scheme as described in Remark C.2.

**Privacy.** The following privacy guarantee of our algorithm is a direct corollary of Lemma C.4.

**Lemma F.7** (Privacy)**.** *Algorithm F.6 is $2\varepsilon$-differentially node private.*

**Utility.** The utility guarantee of our algorithm is stated in the following lemma.

**Lemma F.8** (Utility). *There are constants $C_1, C_2, C_3$ such that the following holds. For any $\eta \leqslant C_1$, $\varepsilon \geqslant C_2 \log(n)/n$, and $d^\circ \geqslant C_3$, given an $\eta$-corrupted Erdős-Rényi random graph $\mathbb{G}(n, d^\circ/n)$ and a coarse estimate $\hat{d}$ such that $0.5d^\circ \leqslant \hat{d} \leqslant 2d^\circ$, Algorithm F.6 outputs an estimate $\tilde{d}$ satisfying*

$$\left| \frac{\tilde{d}}{d^\circ} - 1 \right| \leqslant O\left( \sqrt{\frac{\log n}{d^\circ n}} + \frac{\log^2 n}{\sqrt{d^\circ}\varepsilon n} + \frac{\eta \log n}{\sqrt{d^\circ}} \right),$$

*with probability $1 - n^{-\Omega(1)}$.*

Before proving Lemma F.8, we need the following two lemmas.

**Lemma F.9** (Volume of low-score points). *Let $A \in \mathbb{R}^{n \times n}$ and $\varepsilon > 0$. Consider the distribution $\mu_{A,\varepsilon}$ defined by Eq. (F.18). Suppose $(Y = A^*, z = z^*)$ is a solution to $\mathcal{E}(Y, z; A, \gamma^*)$ and $d(A^*) \geqslant 2$. Then for any $t \geqslant 0$,*

$$\mathop{\mathbb{P}}_{\boldsymbol{d} \sim \mu_{A,\varepsilon}} \left( s(\boldsymbol{d}; A) \geqslant \gamma^* n + \frac{t \log n}{\varepsilon} \right) \leqslant \frac{n^{-t+1}}{\alpha}.$$

*Proof.* Apply Lemma C.5 with $\mathcal{D} = [0, n]$ and

$$\mathcal{G}(A^*) = \left\{ d \in \mathcal{D} : \frac{d(A^*)}{1 + \alpha} \leqslant d \leqslant \frac{d(A^*)}{1 - \alpha} \right\}.$$

As $[d(A^*)/(1 + \alpha), d(A^*)] \subseteq \mathcal{G}(A^*)$ and $d(A^*) \geqslant 2 \geqslant 1 + \alpha$, we have $\mathrm{vol}(\mathcal{G}(A^*)) \geqslant \alpha$. $\qquad\square$

**Lemma F.10** (Low score implies utility). *Let $A \in \mathbb{R}^{n \times n}$ and consider the score function $s(\cdot\,; A)$ defined in Eq. (F.17). Suppose $(Y = A^*, z = z^*)$ is a solution to $\mathcal{E}(Y, z; A, \gamma^*)$. For a scalar $d$ such that $s(d; A) \leqslant \tau n$ and $\gamma^* + \tau \leqslant 0.1$,*

$$\frac{1}{1 + \alpha} \left( d(A^*) - 5(\gamma^* + \tau)(\sigma + \delta)\sqrt{\hat{d}} \right) \leqslant d \leqslant \frac{1}{1 - \alpha} \left( d(A^*) + 5(\gamma^* + \tau)(\sigma + \delta)\sqrt{\hat{d}} \right).$$

*Proof.* Applying Lemma F.2 with $\gamma^* + \tau \leqslant 0.1$, we have

$$\mathcal{E}(Y, z; A, \tau) \left|\frac{Y,z}{8}\right. |d(Y) - d(A^*)| \leqslant 5(\gamma^* + \tau)(\sigma + \delta)\sqrt{\hat{d}}.$$

Thus,

$$\mathcal{E}(Y, z; A, \tau) \cup \{|d(Y) - d| \leqslant \alpha d\}$$
$$\left|\frac{Y,z}{8}\right. \frac{1}{1 + \alpha} \left( d(A^*) - 5(\gamma^* + \tau)(\sigma + \delta)\sqrt{\hat{d}} \right) \leqslant d \leqslant \frac{1}{1 - \alpha} \left( d(A^*) + 5(\gamma^* + \tau)(\sigma + \delta)\sqrt{\hat{d}} \right).$$

$\qquad\square$

Now we are ready to prove Lemma F.8.

*Proof of Lemma F.8.* Let $A$ be a realization of $\eta$-corrupted $\mathbb{G}(n, d^\circ/n)$. By Lemma F.3, the following event happens with probability at least $1 - n^{-\Omega(1)}$. There exists a solution $(Y = A^*, z = z^*)$ to $\mathcal{E}(Y, z; A, \gamma^*, \sigma, \delta, \hat{d})$ where $\gamma^* = \eta$, $\sigma \leqslant O(\log n)$, $\delta \leqslant O(\sqrt{\log n})$, and $|d(A^*) - d^\circ| \leqslant O\left( \sqrt{d^\circ \log(n)/n} \right)$.

As $d(A^*) \geqslant 0.9d^\circ \geqslant 2$, then it follows by setting $t = 10$ and $\alpha = n^{-2}$ in Lemma F.9 that,

$$\mathop{\mathbb{P}}_{\boldsymbol{d} \sim \mu_{A,\varepsilon}} (s(\boldsymbol{d}; A) \leqslant \tau n) \geqslant 1 - n^{-7} \text{ where } \tau := 2\eta + 10 \log(n)/(\varepsilon n).$$

Let $\tilde{d}$ be a scalar such that $s(\tilde{d}; A) \leqslant \tau n$. For $\eta$ and $\log(n)/(\varepsilon n)$ smaller than some constant, we have $2\eta + \tau \leqslant 0.1$. Then by Lemma F.10,

$$\frac{1}{1 + \alpha} \left( d(A^*) - 5(\eta + \tau)(\sigma + \delta)\sqrt{\hat{d}} \right) \leqslant \tilde{d} \leqslant \frac{1}{1 - \alpha} \left( d(A^*) + 5(\eta + \tau)(\sigma + \delta)\sqrt{\hat{d}} \right).$$

Plugging in everything, we have

$$\left| \frac{\tilde{d}}{d^\circ} - 1 \right| \leqslant O\left( \sqrt{\frac{\log n}{d^\circ n}} + \frac{\log^2 n}{\sqrt{d^\circ} \varepsilon n} + \frac{\eta \log n}{\sqrt{d^\circ}} \right).$$

$\square$

**Proof of Theorem F.1.** By Lemma F.7 and Lemma F.8.

## G  Lower bounds

In this section, we prove Theorem 1.5, Theorem 1.7, and Theorem 1.8.

### G.1  Lower bound for Erdős-Rényi random graphs

In this section, we prove Theorem 1.5.

**Theorem** (Restatement of Theorem 1.5)**.** *Suppose there is an $\varepsilon$-differentially node-private algorithm that, given an Erdős-Rényi random graph $\mathbb{G}(n, p^\circ)$, outputs an estimate $\tilde{p}$ satisfying $|\tilde{p}/p^\circ - 1| \leqslant \alpha$ with probability $1 - \beta$. Then we must have*

$$\alpha \geqslant \Omega\left( \frac{\log(1/\beta)}{\varepsilon n \sqrt{n p^\circ}} \right).$$

We leave the formal proof of Theorem 1.5 to the end of this section. Now we sketch the proof idea. One natural idea to prove this theorem is to construct a coupling $\omega$ of $\mathbb{G}(n, p^\circ)$ and $\mathbb{G}(n, (1 - 2\alpha)p^\circ)$ such that for $(G, G') \sim \omega$, the typical distance between $G$ and $G'$ can be well controlled. However, such a coupling is tricky to construct directly, as the node degrees in an Erdős-Rényi random graph are not independent. To avoid dealing with such dependence, we instead consider the directed Erdős-Rényi random graphs, which is inspired by the proof of [AJK+22, Theorem 1.5]. The directed Erdős-Rényi random graph model, denoted by $\tilde{\mathbb{G}}(n, p^\circ)$, is a distribution over $n$-node directed graphs where each edge $(i, j)$ is present with probability $p^\circ$ independently. Since the outdegrees in a directed Erdős-Rényi random graph are i.i.d. Binomial random variables, it is not so difficult to construct a coupling of $\tilde{\mathbb{G}}(n, p^\circ)$ and $\tilde{\mathbb{G}}(n, (1 - 2\alpha)p^\circ)$. Then we can convert such a coupling into a coupling of $\mathbb{G}(n, p^\circ)$ and $\mathbb{G}(n, (1 - 2\alpha)p^\circ)$.

**Lemma G.1** (Coupling)**.** *Let $p^\circ \in [0, 1]$, $\alpha \in [0, 1/2]$, and $p' := (1 - 2\alpha)p^\circ$. There exists a coupling $\omega$ of $\mathbb{G}(n, p^\circ)$ and $\mathbb{G}(n, p')$ with the following property. For $(G, G') \sim \omega$, the distribution of $\mathrm{dist}(G, G')$ is the binomial distribution $\mathrm{Bin}(n, \Delta)$ where $\Delta = \mathrm{TV}(\mathrm{Bin}(n, p^\circ), \mathrm{Bin}(n, p'))$. Moreover, if $p^\circ \leqslant c$ and $\alpha \leqslant c'/\sqrt{n p^\circ}$ for some constants $c, c'$, then $\Delta \lesssim \alpha \sqrt{n p^\circ}$.*

*Proof.* We first show that it suffices to construct a coupling of $\tilde{\mathbb{G}}(n, p^\circ)$ and $\tilde{\mathbb{G}}(n, p')$. For a directed graph $\tilde{G}$, it can be converted into an undirected graph $U(\tilde{G})$ by letting $\{i, j\} \in U(\tilde{G})$ iff $i \leqslant j$ and $(i, j) \in \tilde{G}$. It is easy to see that[10] $\mathrm{dist}(\tilde{G}, \tilde{G}') = \mathrm{dist}(U(\tilde{G}), U(\tilde{G}'))$. Also observe that if $\tilde{G} \sim \tilde{\mathbb{G}}(n, p^\circ)$ then $U(\tilde{G}) \sim \mathbb{G}(n, p^\circ)$. Therefore, a coupling $\tilde{\omega}$ of $\tilde{\mathbb{G}}(n, p^\circ)$ and $\tilde{\mathbb{G}}(n, p')$ can be easily converted in to a coupling $\omega$ of $\mathbb{G}(n, p^\circ)$ and $\mathbb{G}(n, p')$ such that, for $(\tilde{G}, \tilde{G}') \sim \tilde{\omega}$ and $(G, G') \sim \omega$, we have

$$\mathrm{dist}(\tilde{G}, \tilde{G}') \stackrel{\mathrm{d}}{=} \mathrm{dist}(G, G').$$

Now we construct a coupling of $\tilde{\mathbb{G}}(n, p^\circ)$ and $\tilde{\mathbb{G}}(n, p')$. Instead of sampling each edge independently, an equivalent way to sample from $\tilde{\mathbb{G}}(n, p^\circ)$ is as follows. For each $i \in [n]$ :

---

[10]For a directed graph $\tilde{G}$, we define its adjacency matrix $\tilde{A}$ to be $\tilde{A}(i, j) := \mathbb{1}\{(i, j) \in \tilde{G}\}$. The (node) distance between two $n$-node directed graphs $\tilde{G}, \tilde{G}'$, denoted by $\mathrm{dist}(\tilde{G}, \tilde{G}')$, is number of nonzero rows of $\tilde{A} - \tilde{A}'$.

- Sample an outdegree $d \sim \mathrm{Bin}(n, p^\circ)$.

- Sample a uniformly random subset $S \subseteq [n]$ of size $d$. For each $j \in S$, add an edge from $i$ to $j$.

Then it is easy to see there exists a coupling $\tilde{\omega}$ of $\tilde{\mathbb{G}}(n, p^\circ)$ and $\tilde{\mathbb{G}}(n, p')$ such that if $(\tilde{G}, \tilde{G}') \sim \tilde{\omega}$ then $\mathrm{dist}(\tilde{G}, \tilde{G}') \sim \mathrm{Bin}(n, \Delta)$ where

$$\Delta = \mathrm{TV}(\mathrm{Bin}(n, p^\circ), \mathrm{Bin}(n, p')) .$$

We have the following bound on the total variation between binomial distributions (see e.g. [AJ06, Equation (2.15)]). For $0 < p < 1$ and $0 < x < 1 - p$,

$$\mathrm{TV}(\mathrm{Bin}(N, p), \mathrm{Bin}(N, p + x)) \leqslant \frac{\sqrt{e}}{2} \frac{\tau(x)}{(1 - \tau(x))^2} ,$$

where $\tau(x) := x\sqrt{\frac{N+2}{2p(1-p)}}$, provided $\tau(x) < 1$. Plugging in $N = n$, $p = p^\circ$, and $x = 2\alpha p^\circ$, we have

$$\Delta = \mathrm{TV}(\mathrm{Bin}(n, p^\circ), \mathrm{Bin}(n, p')) \lesssim \alpha \sqrt{np^\circ} ,$$

provided $p^\circ \leqslant c$ and $\alpha \leqslant c'/\sqrt{np^\circ}$ for sufficiently small absolute constants $c, c'$. $\qquad \square$

*Proof of Theorem 1.5.* Let $\mathcal{A}$ be an algorithm satisfying the theorem's assumptions. Let $p' := (1 - 2\alpha)p^\circ$. Let $\omega$ be a coupling of $\mathbb{G}(n, p^\circ)$ and $\mathbb{G}(n, p')$ as guaranteed by Lemma G.1. Then for $(G, G') \sim \omega$, we have $\mathrm{dist}(G, G') \sim \mathrm{Bin}(n, \Delta)$ where $\Delta = \mathrm{TV}(\mathrm{Bin}(n, p^\circ), \mathrm{Bin}(n, p'))$.

By the utility assumption of algorithm $\mathcal{A}$,

$$\mathop{\mathbb{P}}_{\mathcal{A}, \mathbb{G}(n,p^\circ)} (|\mathcal{A}(G) - p^\circ| < \alpha p^\circ) \geqslant 1 - \beta .$$

As algorithm $\mathcal{A}$ is $\varepsilon$-DP, we have for any graphs $G, G'$ that,

$$\mathop{\mathbb{P}}_{\mathcal{A}} (|\mathcal{A}(G') - p'| < \alpha p^\circ) \leqslant e^{\varepsilon \cdot \mathrm{dist}(G,G')} \cdot \mathop{\mathbb{P}}_{\mathcal{A}} (|\mathcal{A}(G) - p'| < \alpha p^\circ) .$$

Taking expectation w.r.t. the coupling $\omega$ on both sides gives

$$\mathop{\mathbb{E}}_{\omega} \mathop{\mathbb{E}}_{\mathcal{A}} \mathbb{1}\{|\mathcal{A}(G') - p'| < \alpha p^\circ\} \leqslant \mathop{\mathbb{E}}_{\omega} e^{\varepsilon \cdot \mathrm{dist}(G,G')} \cdot \mathop{\mathbb{E}}_{\mathcal{A}} \mathbb{1}\{|\mathcal{A}(G) - p'| < \alpha p^\circ\} ,$$

$$\mathop{\mathbb{P}}_{\mathcal{A}, \tilde{\mathbb{G}}(n,p')} (|\mathcal{A}(G') - p'| < \alpha p^\circ) \leqslant \mathop{\mathbb{E}}_{\omega, \mathcal{A}} e^{\varepsilon \cdot \mathrm{dist}(G,G')} \cdot \mathbb{1}\{|\mathcal{A}(G) - p'| < \alpha p^\circ\} . \qquad (\mathrm{G.1})$$

By the utility assumption of algorithm $\mathcal{A}$ and $p' < p^\circ$, the left-hand side of Eq. (G.1) is at least $1 - \beta$. Using the Cauchy-Schwartz inequality, the right-hand side of Eq. (G.1) can be upper bounded as follows,

$$\mathop{\mathbb{E}}_{\omega, \mathcal{A}} e^{\varepsilon \cdot \mathrm{dist}(G,G')} \cdot \mathbb{1}\{|\mathcal{A}(G) - p'| < \alpha p^\circ\} \leqslant \sqrt{\mathop{\mathbb{E}}_{\omega, \mathcal{A}} e^{2\varepsilon \cdot \mathrm{dist}(G,G')}} \sqrt{\mathop{\mathbb{E}}_{\omega, \mathcal{A}} \mathbb{1}\{|\mathcal{A}(G) - p'| < \alpha p^\circ\}}$$

$$\leqslant \sqrt{\mathop{\mathbb{E}}_{\mathrm{Bin}(n,\Delta)} e^{2\varepsilon \cdot X}} \sqrt{\mathop{\mathbb{P}}_{\mathcal{A}, \tilde{\mathbb{G}}(n,p^\circ)} (|\mathcal{A}(G) - p'| < \alpha p^\circ)} .$$

By squaring both sides of Eq. (G.1) and plugging in the above two bounds, we have

$$(1 - \beta)^2 \leqslant \mathop{\mathbb{E}}_{\mathrm{Bin}(n,\Delta)} [e^{2\varepsilon \cdot X}] \cdot \mathop{\mathbb{P}}_{\mathcal{A}, \tilde{\mathbb{G}}(n,p^\circ)} (|\mathcal{A}(G) - p'| < \alpha p^\circ) .$$

Using the formula for the moment generating function of binomial distributions, we have

$$\mathop{\mathbb{E}}_{\mathrm{Bin}(n,\Delta)} [e^{2\varepsilon \cdot X}] = \left(1 + \Delta(e^{2\varepsilon} - 1)\right)^n \leqslant e^{n\Delta(e^{2\varepsilon}-1)} .$$

Then

$$\mathop{\mathbb{P}}_{\mathcal{A}, \mathbb{G}(n,p^\circ)} (|\mathcal{A}(G) - p'| < \alpha p^\circ) \geqslant (1 - \beta)^2 \cdot e^{-n\Delta(e^{2\varepsilon}-1)} .$$

Since $p' - p^\circ = 2\alpha p^\circ$, the two events $\{\hat{p} : |\hat{p} - p^\circ| < \alpha p^\circ\}$ and $\{\hat{p} : |\hat{p} - p'| < \alpha p^\circ\}$ are disjoint. Thus,

$$\mathbb{P}_{\mathcal{A}, \mathbb{G}(n, p^\circ)}(|\mathcal{A}(G) - p'| < \alpha p^\circ) \leqslant 1 - \mathbb{P}_{\mathcal{A}, \mathbb{G}(n, p^\circ)}(|\mathcal{A}(G) - p^\circ| < \alpha p^\circ) \leqslant \beta \,.$$

Therefore, we have the following lower bound

$$\Delta \geqslant \frac{2\log(1 - \beta) + \log(1/\beta)}{n(e^{2\varepsilon} - 1)} \,,$$

which is $\Delta \gtrsim \frac{\log(1/\beta)}{n\varepsilon}$ for samll enough $\varepsilon$ and $\beta$.

By Lemma G.1, if $p^\circ \leqslant c$ and $\alpha \leqslant c'/\sqrt{np^\circ}$ for some constants $c, c'$, then $\Delta \lesssim \alpha\sqrt{np^\circ}$. Combined with the lower bound $\Delta \gtrsim \frac{\log(1/\beta)}{n\varepsilon}$, we have

$$\alpha \gtrsim \frac{\log(1/\beta)}{n\varepsilon\sqrt{np^\circ}} \,.$$

$\square$

### G.2 Lower bound for inhomogeneous random graphs

In this section, we prove Theorem 1.7 and Theorem 1.8.

We first show the lower bound for the error rate of robust estimation.

**Theorem** (Restatement of Theorem 1.7). *Suppose there is an algorithm satisfies the following guarantee for any symmetric matrix $Q^\circ \in [0, 1]^{n \times n}$. Given an $\eta$-corrupted inhomogeneous random graph $\mathbb{G}(n, Q^\circ)$, the algorithm outputs an estimate $\hat{p}$ satisfying $|\hat{p}/p^\circ - 1| \leqslant \alpha$ with probability at least $0.99$, where $p^\circ = \sum_{i,j} Q^\circ_{ij}/(n^2 - n)$. Then we must have $\alpha \geqslant \Omega(R\eta)$, where $R = \max_{i,j} Q^\circ_{ij}/p^\circ$.*

*Proof.* Let $p^\circ \in [0, 1]$, and let $Q^\circ \in [0, 1]^{n \times n}$ be the matrix, in which all entries are $p^\circ$, except for the rows and columns corresponding to a set of $\eta n$ nodes setting to be $Rp^\circ$. Let $Q$ be the matrix, in which all entries are $p^\circ$, except for the rows and columns corresponding to a set of $\eta n$ nodes setting to be $0$.

We construct the following pair of distributions $\mathcal{D}_0$ and $\mathcal{D}_1$:

- $\mathcal{D}_0$: The distribution of $G \sim \mathbb{G}(Q^\circ)$.

- $\mathcal{D}_1$: The distribution of $G \sim \mathbb{G}(Q)$.

Then we have $\frac{1}{n^2}|\|Q^\circ\|_1 - \|Q\|_1| \geqslant \Omega(R\eta n^2 p^\circ)$.

On the other hand, there is a coupling between $\tilde{G} \sim \mathbb{G}(Q^\circ)$ and $\tilde{G}' \sim \mathbb{G}(Q)$ such that $\mathrm{dist}(\tilde{G}, \tilde{G}') \leqslant \eta n$. Therefore, the two distributions are indistinguishable under the $\eta$-corruption model. Since the edge density of $\mathbb{G}(Q^\circ)$ differs from $\mathbb{G}(Q)$ by $\Omega(R\eta p^\circ)$, no algorithm can achieve error rate $o(R\eta p^\circ)$ with probability $1 - o(1)$ for both distributions under the corruption of $\eta$-fraction of the nodes. $\square$

**Theorem** (Restatement of Theorem 1.8). *Suppose there is an $\varepsilon$-differentially node-private algorithm satisfies the following guarantee for any symmetric matrix $Q^\circ \in [0, 1]^{n \times n}$. Given an inhomogeneous random graph $\mathbb{G}(n, Q^\circ)$, the algorithm outputs an estimate $\hat{p}$ satisfying $|\hat{p}/p^\circ - 1| \leqslant \alpha$ with probability $1 - \beta$, where $p^\circ = \sum_{i,j} Q^\circ_{ij}/(n^2 - n)$. Then we must have*

$$\alpha \geqslant \Omega\left(\frac{R\log(1/\beta)}{n\varepsilon}\right) \,,$$

*where $R = \max_{i,j} Q^\circ_{ij}/p^\circ$.*

*Proof of Theorem 1.8.* We will prove the lower bound by constructing a pair of distributions $\mathcal{D}_0$ and $\mathcal{D}_1$ such that the total variation distance between them is small, but the difference in edge density is significant. Then since $\varepsilon$-differentially node-private algorithm needs to have similar distributions in the output, it could not succeed in accurately estimating the edge density accurately under both distributions.

Let $\eta \in [0, 0.001)$. Let $p^\circ \in [0, 1]$, and let $Q^\circ \in [0, 1]^{n \times n}$ be the matrix, in which all entries are $p^\circ$, except for the rows and columns corresponding to a set of $\eta n$ nodes setting to be $0$. Let $Q$ be the matrix, in which all entries are $p^\circ$, except for the rows and columns corresponding to a set of $\eta n$ nodes setting to be $Rp^\circ$.

We construct the following pair of distributions $\mathcal{D}_0$ and $\mathcal{D}_1$:

- $\mathcal{D}_0$: The distribution of $G \sim \mathbb{G}(Q^\circ)$.

- $\mathcal{D}_1$: The distribution of $G \sim \mathbb{G}(Q)$.

Let $p' = \frac{\|Q^\circ\|_1}{n^2}$ and $p = \frac{\|Q\|_1}{n^2}$. We have $|p - p'| \geqslant R\eta p^\circ$.

On the other hand, there is a coupling between $\tilde{G} \sim \mathbb{G}(Q)$ and $\tilde{G}' \sim \mathbb{G}(Q^\circ)$ such that $\mathrm{dist}(\tilde{G}, \tilde{G}') \leqslant \eta n$. Taking expectation w.r.t. the coupling $\omega$ on both sides gives

$$\mathop{\mathbb{E}}_{\omega} \mathop{\mathbb{E}}_{\mathcal{A}} \mathbb{1}\left\{\left|\mathcal{A}(\tilde{G}') - p\right| < \frac{R\eta}{2}p^\circ\right\} \leqslant \mathop{\mathbb{E}}_{\omega} e^{\varepsilon \cdot \mathrm{dist}(\tilde{G}, \tilde{G}')} \cdot \mathop{\mathbb{E}}_{\mathcal{A}} \mathbb{1}\left\{\left|\mathcal{A}(\tilde{G}) - p\right| < \frac{R\eta}{2}p^\circ\right\},$$

$$\mathop{\mathbb{P}}_{\mathcal{A}, \tilde{G}(Q^\circ)}\left(\left|\mathcal{A}(\tilde{G}') - p\right| < \frac{R\eta}{2}p^\circ\right) \leqslant \mathop{\mathbb{E}}_{\omega, \mathcal{A}} e^{\varepsilon \cdot \mathrm{dist}(\tilde{G}, \tilde{G}')} \cdot \mathbb{1}\left\{\left|\mathcal{A}(\tilde{G}) - p\right| < \frac{R\eta}{2}p^\circ\right\}.$$

By the utility assumption of algorithm $\mathcal{A}$ and $p < p^\circ$, the left-hand side is at least $1 - \beta$. Using the Cauchy-Schwartz inequality, the right-hand side can be upper bounded as follows,

$$\mathop{\mathbb{E}}_{\omega, \mathcal{A}} e^{\varepsilon \cdot \mathrm{dist}(\tilde{G}, \tilde{G}')} \cdot \mathbb{1}\left\{\left|\mathcal{A}(\tilde{G}) - p\right| < \frac{R\eta}{2}p^\circ\right\} \leqslant \sqrt{\mathop{\mathbb{E}}_{\omega, \mathcal{A}} e^{2\varepsilon \cdot \mathrm{dist}(\tilde{G}, \tilde{G}')}} \sqrt{\mathop{\mathbb{E}}_{\omega, \mathcal{A}} \mathbb{1}\left\{\left|\mathcal{A}(\tilde{G}) - p\right| < \frac{R\eta}{2}p^\circ\right\}}$$

$$\leqslant \exp(\varepsilon \eta n) \sqrt{\mathop{\mathbb{P}}_{\mathcal{A}, \tilde{G}(Q)}\left(\left|\mathcal{A}(\tilde{G}) - p\right| < \frac{R\eta}{2}p^\circ\right)}.$$

Thus

$$(1 - \beta)^2 \leqslant \exp(\varepsilon \eta n) \cdot \mathop{\mathbb{P}}_{\mathcal{A}, \tilde{G}(Q)}\left(\left|\mathcal{A}(\tilde{G}) - p\right| < \frac{R\eta}{2}p^\circ\right).$$

Then

$$\mathop{\mathbb{P}}_{\mathcal{A}, \tilde{G}(Q)}\left(\left|\mathcal{A}(\tilde{G}) - p\right| < \frac{R\eta}{2}p^\circ\right) \geqslant (1 - \beta)^2 \cdot \exp(-\varepsilon \eta n).$$

Since $|p - p^\circ| \geqslant R\eta p^\circ$, the two events $\{\hat{p} : |\hat{p} - p^\circ| < \frac{R\eta}{2}p^\circ\}$ and $\{\hat{p} : |\hat{p} - p| < \frac{R\eta}{2}p^\circ\}$ are disjoint, which implies

$$\mathop{\mathbb{P}}_{\mathcal{A}, \tilde{G}(Q)}\left(\left|\mathcal{A}(\tilde{G}) - p\right| < \frac{R\eta}{2}p^\circ\right) \leqslant 1 - \mathop{\mathbb{P}}_{\mathcal{A}, \tilde{G}(Q)}\left(\left|\mathcal{A}(\tilde{G}) - p^\circ\right| < \frac{R\eta}{2}p^\circ\right) \leqslant \beta.$$

Therefore, we have $\beta \geqslant (1 - \beta)^2 \exp(-\varepsilon \eta n)$. As result, we need to have $\eta \geqslant \Omega\left(\frac{\log(\beta)}{\varepsilon n}\right)$. Thus we have

$$|p - p'| \geqslant \Omega\left(\frac{R \log(\beta)p^\circ}{\varepsilon n}\right).$$

Since $p^\circ \geqslant p'$, it follows that

$$|p - p'| \geqslant \Omega\left(\frac{R \log(\beta)p'}{\varepsilon n}\right),$$

which finishes the proof. $\square$

