# OpenReview forum: "Private Edge Density Estimation for Random Graphs: Optimal, Efficient and Robust"
_NeurIPS.cc/2024/Conference — NeurIPS 2024 spotlight_

### Official Review · Reviewer_97Rf · 2024-06-28

**Soundness:** 3
**Presentation:** 3
**Contribution:** 2
**Rating:** 6
**Confidence:** 3

**Summary:**

This paper studies edge density estimation for Erdos-Renyi random graph under node-level differential privacy. They then show that their approach can actually be used to estimating edge density for inhomogeneous graphs. In particular, this paper proposes an efficient algorithm with optimal privacy cost $O(1/(\varepsilon n \sqrt{np}))$, which is negligible to the non-private error for any epsilon in a moderate regime. To achieve this, the main technical approach in this paper is to design a new sum-of-squares algorithm for robust edge density estimation, and then based on the reduction from privacy to robustness in Hopkins et al., this paper uses an exponential mechanism whose score function is based on the correspoinding the sum-of-squares program.

To design a robust algorithm for edge density estimation of ER random graphs, this paper established several polynomial constraints to ensure that a ER random graph will satisfy with high probability. The proof that if a graph meets these constraints, its average degree will remain close to $d$ is straightforward enough for the sum-of-squares proof system, extends the utility guarantee of the polynomial program to its semidefinite programming relaxation, resulting in a polynomial-time robust algorithm.

**Strengths:**

1. Graph parameter approximation under node-level privacy is an important topic, and this paper gives optimal algorithm with a polynomial time implementation by sum-of-squares relaxation, under the most natrual setting of random graphs.
2. Their techniques of using sum-of-squares method for exponential mechanism is sophisticated.
3. All theorems are clearly stated and technically correct.

**Weaknesses:**

As far as I can see, the most relevant work is Chen et al. "Private graphon estimation via sum-of-squares", but it appears to me that this paper lacks discussion and comparison in [Chen et al.]. Please see "Questions" for details. I would be happy to raise my score if all my questions are adressed.

**Questions:**

1. Could you please elaborate on which part of your proof technique deviates from that in [Chen et al.]? Alternatively, are the proof techniques similar, while you are just addressing different problems in this paper?
2. There is also a theorem in [Chen et al.] for edge density estimation of stochastic block random graphs (Lemma 4.10), and it appears that their utility is better. Could you explain the fundamental obstacle between stochastic block random graphs and the more general inhomogeneous graphs that prevents the estimation for inhomogeneous graphs from achieving results as good as those for stochastic block random graphs? Would you mind adding more discussion about the results in [Chen et al.]?

**Limitations:**

This paper discusses several limitations.

---

> ### Author Rebuttal · Authors · 2024-08-06
>
> **Question 1:**
> > Could you please elaborate on which part of your proof technique deviates from that in [Chen et al.]? Alternatively, are the proof techniques similar, while you are just addressing different problems in this paper?
>
> **Response 1:**
> On the one hand, the algorithm in [CDd+24] uses an edge density estimation algorithm based on [SU19] as a preprocessing step and proceeds by an sum-of-square exponential mechanism to privately estimate the underlying graphon.
> Concretely for the task of edge density estimation, [CDd+24] does not improve over [SU19] and uses in fact the same algorithm as [SU19].
> To improve the edge density estimation results in [SU19], we developed a new private edge density estimation algorithm using the sum-of-squares hierarchy, which is completely different from the techniques in [SU19] based on smooth sensitivity.
>
> On the other hand, although [CDd+24] also relies on the sum-of-squares exponential mechanism for graphon estimation, our program constraints and identifiability proof are significantly different.
> There are two particularly notable technical differences:
>
> - Our strategy is closely related to the reduction from robustness to privacy framework developed in [HKMN23], while [CDd+24] relies on sum-of-squares Lipshitz extensions.
> - The private graphon estimation algorithm in [CDd+24] relies on a single sum-of-squares program, while our algorithm has two stages: the rough estimation and the fine estimation, each with a different sum-of-squares program. The reason is that we found it difficult to achieve nearly optimal error rate using a single sum-of-squares program.
>
> **Question 2:**
> > There is also a theorem in [Chen et al.] for edge density estimation of stochastic block random graphs (Lemma 4.10), and it appears that their utility is better. Could you explain the fundamental obstacle between stochastic block random graphs and the more general inhomogeneous graphs that prevents the estimation for inhomogeneous graphs from achieving results as good as those for stochastic block random graphs? Would you mind adding more discussion about the results in [Chen et al.]?
>
> **Response 2:**
> We will add more discussion about the results in [CDd+24] in our proceedings version.
> For the task of edge density estimation, [CDd+24] actually does not make use of any structure specific to stochastic block random graphs, but just treat them as inhomogeneous random graphs.
> So [CDd+24, Lemma 4.10] is essentially a result on edge density estimation of inhomogeneous random graphs.
> The algorithm behind [CDd+24, Lemma 4.10] uses an edge density estimation algorithm based on [SU19].
> The error bound stated in [CDd+24, Lemma 4.10] is actually only the privacy cost. The total error bound of [CDd+24, Lemma 4.10] should also include the non-private error which is the same as the non-private error in our Theorem 1.6.
> In terms of privacy cost, our Theorem 1.6 actually improves over the guarantees of Lemma 4.10 in [CDd+24].
> More specifically, the privacy cost of [CDd+24, Lemma 4.10] is
> $$
>     |{\hat{\rho}-\rho}|^2 \leq \tilde{O} \left(\frac{R^2\rho^2}{\epsilon^2 n^2}+\frac{1}{\epsilon^4 n^4}\right) .
> $$
> In comparison, our Theorem 1.6 gives a privacy cost of
> $$
>     |{\hat{\rho}-\rho}|^2 \leq \tilde{O} \left(\frac{R^2\rho^2}{\epsilon^2 n^2}\right) .
> $$
>
>
> **Reference:**
>
> - [CDd+24] Chen, Hongjie, et al. "Private graphon estimation via sum-of-squares." Proceedings of the 56th Annual ACM Symposium on Theory of Computing. 2024.
> - [SU19] Ullman, Jonathan, and Adam Sealfon. "Efficiently estimating erdos-renyi graphs with node differential privacy." Advances in Neural Information Processing Systems 32 (2019).
> - [HKMN23] Hopkins, Samuel B., et al. "Robustness implies privacy in statistical estimation." Proceedings of the 55th Annual ACM Symposium on Theory of Computing. 2023.

---

> > ### Comment · Reviewer_97Rf · 2024-08-11
> >
> > Thank you for the rebuttal.

---

### Official Review · Reviewer_Se7o · 2024-07-12

**Soundness:** 3
**Presentation:** 3
**Contribution:** 3
**Rating:** 7
**Confidence:** 3

**Summary:**

The authors propose a robust and efficient (polynomial time) algorithm to estimate the edge density of Erdos-Renyi graph, which achieves optimal error up to logarithmic factors. The authors also design an optimal algorithm for inhomogenous graphs where edges are drawn iid from different Bern(p_i).

**Strengths:**

1. The theoretical results are very strong. The authors prove matching (up to logarithmic factors) error bounds for both Inhomogeneous and Erdos-Renyi graphs. The algorithms are polynomial-time.
2. The techniques are interesting. The authors use recent results for the connection between privacy and robustness. To design a time-efficient algorithm, the authors go beyond simple reduction from privacy to robustness and apply the sum-of-squares method.
3. The paper is well-written.

**Weaknesses:**

There is no obvious weakness with regard to the results and technical parts of the paper. The only minor weaknesses are the problem formulation.
1. Both the Erdos-Renyi graph and the inhomogeneous model assume edges are independently chosen. In real applications, graphs are often fixed, and edges may not be independent.
2. The algorithm only works for edge density estimation. There should be many other interesting graph statistics, even on the Erdos-Renyi graph.

**Questions:**

1. Is it possible to design algorithms when the entries of the connection probability matrix are not independent?
2. Do any algorithmic ideas translate to other graph statistics like the number of triangles and k-stars?

**Limitations:**

Limitations and negative social impact are adequately addressed.

---

> ### Author Rebuttal · Authors · 2024-08-06
>
> **Question 1:**
> > Is it possible to design algorithms when the entries of the connection probability matrix are not independent?
>
> **Response 1:**
> Yes, our algorithm guarantees can extend to some settings where the edges in the graph are not independent:
>
> - As our algorithm is robust under node corruption, the guarantees still hold even under minor dependence between the edges in the graph.
> - Moreover, we expect that the guarantees of our algorithm easily extend to graphs with bounded maximum degree and bounded spectral norm, as we only exploit these properties of the graph for adding constraints to the sum-of-squares semidefinite programming.
>
> These assumptions are easy to test given the graph instance, and are significantly weaker than the independence assumption.
>
>
> **Question 2:**
> > Do any algorithmic ideas translate to other graph statistics like the number of triangles and k-stars?
>
> **Response 2:**
> This is a great question.
>
> - For Erdos-Renyi random graphs, as the only parameter for the distribution is the edge density $p^\circ$, we can obtain non-trivial (and potentially optimal) guarantees for estimating the number of triangles/k-stars by relating the expectation to $p$.
> - For inhomogenous random graphs, we are not so sure. We expect our techniques can lead to private algorithms for counting constant-size subgraphs such as triangles and k-stars with non-trivial guarantees, when the inhomogeneity (measured by the ratio between the maximum edge connection probability and the average edge density) is bounded.
> The observation is that, when a small $\eta$-fraction of the nodes are corrupted, the number of triangles/k-stars in a large induced subgraph of the given corrupted graph with bounded degree and supported on $(1-\eta)n$ vertices is still close to the total number of triangles/k-stars in the original uncorrupted random graph.

---

> > ### Comment · Reviewer_Se7o · 2024-08-12
> > **Thanks for your response**
> >
> > Thanks for addressing my questions. I am keeping my score.

---

### Official Review · Reviewer_teKp · 2024-07-13

**Soundness:** 3
**Presentation:** 3
**Contribution:** 3
**Rating:** 7
**Confidence:** 2

**Summary:**

This paper gives the first polynomial time DP algorithm for estimating edge density of random graphs (Erdos-Renyi and inhomogeneous). The authors also give information-theoretic lower bounds to show that the error achieved is optimal (up to log factors). The paper utilizes the recent results of Hopkins et al who gave a black box reduction from privacy to robustness via a sum-of-squares exponential mechanism to design their new algorithm. Their main new contribution is a sum-of-squares algorithm for estimating the edge density which they then use together with the Hopkins et al framework to get their final algorithm.

**Strengths:**

The main result of the paper is novel and significantly advances the area of DP algorithms for estimating graph parameters in random graphs.

**Weaknesses:**

The techniques used heavily rely on the recent framework given by the Hopkins et al STOC paper. What is unclear to me is --- what were the challenges in designing the sum-of-squares algorithm for estimating the edge density? Since this algorithm is the main contribution of the paper in some sense, I would like to understand better if the design of the sum-of-squares algorithm itself was previously known and one only needed the Hopkins et al framework to obtain this result.
General comment about the paper --- you should define the quantity edge density formally somewhere since the whole paper hinges on the reader understanding what it is.

**Questions:**

The results of this paper are very important and exciting. However, can you expand on the challenges in designing the sum-of-squares algorithm for estimating the edge density? I would like to be more convinced that this result is not simply piggybacking off on the Hopkins et al result. Thanks!

**Limitations:**

There was no explicit discussion regarding the limitations of this work in the main body.

---

> ### Author Rebuttal · Authors · 2024-08-06
>
> **Comment 1:**
> > You should define the quantity edge density formally somewhere since the whole paper hinges on the reader understanding what it is.
>
> **Response 1:**
> Thank you for pointing it out! We will add a formal definition in our proceedings version.
>
>
> **Question 2:**
> > Can you expand on the challenges in designing the sum-of-squares algorithm for estimating the edge density? I would like to be more convinced that this result is not simply piggybacking off on the Hopkins et al result.
>
> **Response 2:**
> This is a great question. We will discuss in more detail in the proceedings version.
> - [HKMN23] and [AUZ23] showed a general connection between privacy and robustness. In general, this connection does not provide guarantees in terms of computational complexity. However there are several important examples where this connection can be used to transform efficient robust algorithms to efficient private algorithms.
> - For the problem of robust edge density estimation under node corruption, there is no known sum-of-squares algorithm before our work, and we are only aware of an iterative algorithm [AJK+22]. For such algorithms not based on convex relaxation, it is completely unclear how to use the aforementioned connection between privacy and robustness estimation toward an efficient private algorithm.
> - We cannot just apply previous robust mean estimation algorithms based on sum-of-squares (e.g. Hopkins et al 2024). On the one hand, if we view edge density estimation as a one-dimensional Bernoulli mean estimation task, then previous algorithms are only optimal under edge corruption, but suboptimal for the node corruption model. On the other hand, if we view edge density estimation as an $n$-dimensional mean estimation task, then samples (i.e. rows of the adjacency matrix) are not independent.
> - In general, when designing a sum-of-squares algorithm, the main challenges are identifying the right set of polynomial constraints and coming up with sum-of-squares proofs.
>
>
> **Reference:**
> - [AJK+22] Acharya, Jayadev, et al. "Robust estimation for random graphs." *Conference on Learning Theory*. PMLR, 2022.
> - [HKMN23] Hopkins, Samuel B., et al. "Robustness implies privacy in statistical estimation." *Proceedings of the 55th Annual ACM Symposium on Theory of Computing*. 2023.
> - [AUZ23] Asi, Hilal, Jonathan Ullman, and Lydia Zakynthinou. "From robustness to privacy and back." *International Conference on Machine Learning*. PMLR, 2023.

---

> > ### Comment · Reviewer_teKp · 2024-08-09
> >
> > Thanks for clarifying, I have updated my score.

---

### Official Review · Reviewer_ZgW7 · 2024-07-15

**Soundness:** 4
**Presentation:** 4
**Contribution:** 4
**Rating:** 8
**Confidence:** 3

**Summary:**

This paper presents a sum-of-squares-based polynomial-time differentially node-private algorithm for estimating the edge density of Erdos-Renyi random graphs that achieves optimal accuracy; the algorithm is simultaneously robust to corruptions. Specifically, if p is the true Erdos-Renyi parameter, then the estimate q released by the algorithm has error |q/p - 1| bounded by 1/(n sqrt{p}) + 1/(eps n \sqrt{np}) + eta / sqrt{np} (up to factors of log n), where eta is the corruption rate for robustness. Up to log n factors, the first term is the information-theoretically optimal error rate with neither privacy or robustness (which is also the error of the empirical edge density), and the third term is known to be required for robustness ([AJK+22]). The paper further shows a lower bound demonstrating that the remaining error term is necessary for privacy (a lower bound had been shown by BCSZ18 for a closely related family of random graphs but not for standard Erdos-Renyi graphs, so the new lower bound closes this gap). Moreover, the paper extend this result to present an algorithm for the much more general problem of edge density estimation in inhomogeneous random graphs (e.g. SBMs or graphons) and proves lower bounds in this setting as well showing that this algorithm also achieves optimal error rate up to logarithmic factors.

**Strengths:**

This paper achieves optimal accuracy in polynomial time for both private and robust edge density estimation of Erdos-Renyi graphs and for the more general setting of inhomogeneous random graphs, nicely closing the gaps from prior work. It achieves this by bringing a new approach to the problem: the sum-of-squares framework, along with leveraging a reduction from privacy to robustness. The paper is generally well-written.

For DP edge density estimation of Erdos-Renyi graphs (even without requiring robustness), the new algorithm matches the (already optimal) accuracy achieved by [BCSZ18] while running in polynomial time instead of exponential time; it also improves over the rate of the polynomial-time algorithm of [SU19], which was suboptimal in the sparse or very-high-privacy regimes (that is, when epsilon * sqrt{pn} << 1). For robust edge density estimation of Erdos-Renyi graphs (even without requiring privacy), the new algorithm also improves of the accuracy of prior work ([AJK+22]) in the sparse regime. Thus, the Erdos-Renyi result improves on SOTA for both privacy and robustness, achieving the optimal bound (up to log n factors). Moreover, the results are extended to the more challenging inhomogenous random graph setting, achieving optimal rate in polynomial time here as well, and novel, tight lower bounds are shown for both settings as well.

**Weaknesses:**

The discussion of prior work is clear for the Erdos-Renyi setting, but it would be useful to have a clearer discussion of what was previously known for the more general inhomogeneous random graph setting under DP and under robustness; see the specific questions 1--2 below.

**Questions:**

1) How does the rate achieved for inhomogeneous random graphs compare to the rate achieved by the exponential-time algorithm of [BCSZ18] for graphons?

2) Is there prior work on robust estimation of general inhomogeneous random graphs, and if so, what rate was achieved?

3) Can we hope to extend the privacy lower bounds (Theorem 1.5 and 1.8) to (epsilon, delta)-DP and not just epsilon-DP?

Minor comment:
line 188: should d^\circ be defined as (n-1) p^\circ instead of n p^\circ?

**Limitations:**

Yes, adequately addressed.

---

> ### Author Rebuttal · Authors · 2024-08-06
>
> **Question 1:**
> > How does the rate achieved for inhomogeneous random graphs compare to the rate achieved by the exponential-time algorithm of [BCSZ18] for graphons?
>
> **Response 1:**
> [BCSZ18] uses the Laplace mechanism for edge density estimation of graphons.
> The privacy cost of their algorithm is $\mathrm{polylog}(n) R/(\epsilon d)$, significantly worse than $\mathrm{polylog}(n) R/(\epsilon n)$ in our result.
>
>
> **Question 2:**
> > Is there prior work on robust estimation of general inhomogeneous random graphs, and if so, what rate was achieved?
>
> **Response 2:**
> To the best of our knowledge, no previous work had studied robust estimation of general inhomogeneous random graphs.
>
>
> **Question 3:**
> > Can we hope to extend the privacy lower bounds (Theorem 1.5 and 1.8) to (epsilon, delta)-DP and not just epsilon-DP?
>
> **Response 3:**
> Our proof of the $\epsilon$-DP lower bound uses a packing argument. Packing arguments can also be used to prove ($\epsilon$,$\delta$)-DP lower bounds, but they usually would not result in meaningful ($\epsilon$,$\delta$)-DP lower bounds.
> A standard and powerful tool to prove ($\epsilon$,$\delta$)-DP lower bounds is the so-called fingerprinting technique [BUV14], which is totally different from packing arguments.
> We expect some non-trivial work is needed to prove meaningful ($\epsilon$,$\delta$)-DP lower bounds for this problem.
> This is an interesting open question.
>
>
> **Comment 4:**
> > line 188: should $d^\circ$ be defined as $(n-1) p^\circ$ instead of $n p^\circ$?
>
> **Response 4:**
> Thank you for pointing this out.
> We define $d^\circ = n p^\circ$ just for notational convenience.
> Strictly speaking, the expected average degree should be $(n-1) p^\circ$ instead of $d^\circ$.
> However, these two quantities are equivalent as $(n-1) p^\circ = \frac{n-1}{n} d^\circ$.
>
> **Reference:**
> - [BUV14] Bun, Mark, Jonathan Ullman, and Salil Vadhan. "Fingerprinting codes and the price of approximate differential privacy." Proceedings of the forty-sixth annual ACM symposium on Theory of computing. 2014.
> - [BCSZ18] Borgs, Christian, et al. "Revealing network structure, confidentially: Improved rates for node-private graphon estimation." *2018 IEEE 59th Annual Symposium on Foundations of Computer Science (FOCS)*. IEEE, 2018.

---

> > ### Comment · Reviewer_ZgW7 · 2024-08-12
> >
> > Thank you for your responses to my questions.

---

### Official Review · Reviewer_ikor · 2024-07-17

**Soundness:** 4
**Presentation:** 3
**Contribution:** 3
**Rating:** 6
**Confidence:** 3

**Summary:**

The paper concerns the graph density estimation question for random Erdos-Renyi graphs. The optimal algorithm, without additional constraints, for this problem is to output the density of edges in the graph.

The paper considers a node-private robust flavor of the question, however:

1. The output is not supposed to reveal much information about any node. More precisely, for any node, the distribution on estimates the algorithm produces is not supposed to be significantly different whether it is included or not. (This task is much easier to achieve in the edge privacy model.)

2. The algorithm is supposed to be robust, i.e., if the edges involving a small number of vertices are arbitrarily edited, this should not throw the estimate off significantly. (This may require, for instance, disregarding some fraction of vertices that are significantly off.)

The paper gives a near optimal algorithm in this setting that runs in polynomial time and it establishes asymptotic bounds on the error achieved by the algorithm.

Additionally, the paper gives an extension of the result to non-homogeneous random graphs.

**Strengths:**

This is a technically impressive achievement with the paper combining many desirable properties of algorithms for a natural toy problem. This is an interesting result for anyone studying private estimation and private release of graph parameters.

**Weaknesses:**

This is not directly a very practical problem, since Erdos-Renyi graphs do not appear frequently in natural contexts and even the generalization assumes that the selection of different edges is independent, which is not true for many models of real-world graphs.

The algorithm may be difficult to implement and run in practice due to the complexity of tools used, which include semidefinite programming.

**Questions:**

Just to make sure, in the inhomogeneous setting, the algorithm has to know $Q^\circ$, right? Otherwise, it could be a zero-one matrix equal to the graph adjacency matrix. What guarantees can be given if $Q^\circ$ is unknown?

**Limitations:**

No limitations.

---

> ### Author Rebuttal · Authors · 2024-08-06
>
> **Comment 1:**
> > This is not directly a very practical problem, since Erdos-Renyi graphs do not appear frequently in natural contexts and even the generalization assumes that the selection of different edges is independent, which is not true for many models of real-world graphs.
>
> **Response 1:**
> Indeed, developing realistic and mathematically tractable random graph models is an important and long-standing question in the field.
> Our algorithm and theoretical analysis extend to more general graphs in the following aspects:
>
> - In the paper, we extend our algorithms and theoretical guarantees for the much more general inhomogeneous random graph models, where we make minimal assumptions beyond the independence between the edges.
> Also, since our algorithm is robust under node corruption, the guarantees will still hold even under minor dependence between the edges in the graph.
>
> - Moreover, we expect that the guarantees of our algorithm easily extend to graphs with bounded maximum degree and bounded spectral radius, which goes beyond the setting where the edges in the graph are independent.
> Particularly these assumptions are easy to test given the graph instance.
>
> Therefore, our work can be viewed as a first step towards private and efficient algorithms for statistical estimation in more realistic graph models.
>
>
> **Comment 2:**
> > The algorithm may be difficult to implement and run in practice due to the complexity of tools used, which include semidefinite programming.
>
> **Response 2:**
> We completely agree that it remains a fascinating and important open question to obtain algorithms for this problem that are more practical or at least have better running times, ideally nearly-linear time.
> Our theoretical work shows that polynomial-time algorithms for this problem exist and that there are no complexity-theoretic obstacles toward practical algorithms.
> We believe that such basic, polynomial time solvable problems ought to have practical algorithms.
> We hope that some of the ideas behind our algorithm could pave the way toward such practical algorithms for this basic problem.
>
>
> **Question 3:**
> > Just to make sure, in the inhomogeneous setting, the algorithm has to know $Q^\circ$, right? Otherwise, it could be a zero-one matrix equal to the graph adjacency matrix. What guarantees can be given if $Q^\circ$ is unknown?
>
>
> **Response 3:**
> No, our algorithm does not need to know $Q^{\circ}$. We assume there is an unknown $Q^{\circ}$. Given a graph generated by $Q^{\circ}$, our goal is to estimate the average of the entries in the matrix $Q^{\circ}$.
> The privacy cost of our algorithm is $\frac{R \log n}{\epsilon n}$ where $R$ is the ratio between the largest entry of $Q^{\circ}$ and the average of $Q^{\circ}$.
> For $Q^{\circ}$ equal to a graph adjacency matrix, we have $R=1/p$ where $p$ is the edge density of the graph.

---

> > ### Comment · Reviewer_ikor · 2024-08-12
> >
> > Thank you for your response and in particular, replying to my question.

---

### Author Rebuttal · Authors · 2024-08-06

We are very grateful to all reviewers for constructive feedback.
We will incorporate these helpful suggestions into the proceedings version of our paper.

**Question:**
What is the key conceptual contribution of our paper, compared to previous work [BCSZ18] and [SU19]?

**Answer:**
[BCSZ18], [SU19] and our work all consider $\epsilon$-differentially node private algorithms for edge density estimation under Erdos-Renyi graph distribution $\mathbb{G}(n,d^\circ/n)$.

- The privacy cost of [SU19] is negligible when $\epsilon \gg (nd^\circ)^{-1/4}$. In comparison, the privacy cost for our algorithm is negligible when $\epsilon \gg n^{-1/2}$, which is a significantly wider privacy parameter range. Our guarantee matches that of the exponential-time algorithm in [BCSZ18].

- [BCSZ18], [SU19] and our work use different techniques. [BCSZ18] uses a generic Lipschitz extension based on exponential mechanisms. [SU19] uses the smooth sensitivity technique. Our work uses the sum-of-squares exponential mechanism.

- [SU19] can only exploit the degree concentration property of Erdos-Renyi graphs. In fact, the privacy cost of [SU19] is optimal on degree-concentrated graphs and thus cannot be further reduced. The reason why our algorithm can surpass their lower bound is that our algorithm can also make use of spectral norm. More specifically, we exploit the property of Erdos-Renyi graphs that the spectral norm of centered adjacency matrix is bounded by $\tilde{O}(\sqrt{d})$. Note for graphs with maximum degree $d$, the centered adjacency matrix can have spectral norm as large as $\Omega(d)$.

- Moreover, we show that the privacy cost of our algorithm is information-theoretically necessary (up to a $\log n$ factor). In this sense, our result is nearly optimal.

- We also extend our results to the inhomogenous random graph models, improving the results from [CDd+24].

**Reference:**

- [BCSZ18] Borgs, Christian, et al. "Revealing network structure, confidentially: Improved rates for node-private graphon estimation." 2018 IEEE 59th Annual Symposium on Foundations of Computer Science (FOCS). IEEE, 2018.
- [SU19] Ullman, Jonathan, and Adam Sealfon. "Efficiently estimating erdos-renyi graphs with node differential privacy." Advances in Neural Information Processing Systems 32 (2019).
- [CDd+24] Chen, Hongjie, et al. "Private graphon estimation via sum-of-squares." Proceedings of the 56th Annual ACM Symposium on Theory of Computing. 2024.

---

### Decision · Program_Chairs · 2024-09-25

**Decision:**

Accept (spotlight)

**Comment:**

This paper considers the problem of differentially private (DP) density estimation for Erdos-Renyi random graphs. The input is an Erdos-Renyi random graph, where each edge is present independently with probability an unknown $p^o$. The goal is to output an estimate to $p^o$. The DP requirement here is w.r.t. node adjacency level, i.e. two graphs are neighbors if they differ only on edges of a single node. The main contribution of this paper is to give a polynomial-time algorithm for this problem that is optimal (up to polylogarithmic factor); previously, only inefficient algorithm [Borgs et al., 2018] or efficient but non-optimal algorithm [Sealfon and Ullman, 2019] are known. In fact, the paper presents an algorithm even in the inhomogeneous case where the probabilities of all pairs may not be equal (but still not too different) and in the "robust" setting where up to some fraction of the nodes can be arbitrarily corrupted. Interestingly, the algorithm presented in this paper provides new results even without privacy constraint in the sparse and highly-corrupted regime. As for the techniques, the high-level algorithm follows those of [Hopkins et al., 2023] in using Sum-of-Square (SoS) relaxation together with exponential mechanism. The main novelty here is in giving a constant-degree SoS proof for identifying $p^o$ robustly.

Overall, this is a technically solid and well-written paper that tackles a fundamental problem and gives nearly tight bounds. The fact that the algorithm provides improvement even in the non-private setting suggests that the techniques can be useful even beyond the privacy context. Given this, we recommend acceptance (spotlight).